



# Regionally optimized high resolution input datasets enhance the representation of snow cover and ecophysiological processes in CLM5

Johanna Teresa Malle[1,2], Giulia Mazzotti[2,3], Dirk Nikolaus Karger[1+], and Tobias Jonas[2+]

[1]Swiss Federal Research Institute for Forest, Snow, and Landscape Research (WSL), Zürcherstrasse 111, 8903, Birmensdorf, Switzerland
[2]WSL Institute for Snow and Avalanche Research SLF, Davos Dorf, Switzerland
[3]Univ. Grenoble Alpes, Université de Toulouse, Météo-France, CNRS, CNRM, Centre d'Études de la Neige, 38100 St. Martin d'Hères, France
[+]Authors contributed equally

**Correspondence:** Johanna Malle (johanna.malle@wsl.ch)

**Abstract.** Land surface processes, crucial for exchanging carbon, nitrogen, water, and energy between the atmosphere and terrestrial Earth, significantly impact the climate system. Many of these processes vary considerably at small spatial and temporal scales, in particular in mountainous terrain and complex topography. To examine the impact of spatial resolution and quality of input data on modeled land surface processes, we conducted simulations using the Community Land Model 5 (CLM5) at

different resolutions and based on a range of input datasets over the spatial extent of Switzerland. Using high-resolution meteorological forcing and land-use data, we found that increased resolution not only improved the representation of snow cover in CLM5 (up to 52% enhancement) but also propagated through the model, directly affecting gross primary productivity and evapotranspiration. These findings highlight the significance of high spatial resolution and high-confidence input datasets in land surface models, enabling better quantification and constraint of process uncertainties. They have profound implications

for climate impact studies. As improvements were observed across the cascade of dependencies in the land surface model, high spatial resolution as well as high-quality forcing data becomes necessary for accurately capturing the impacts of recent climate change. This study further highlights the utility of multi-resolution modeling experiments when aiming to improve process-based representation of variables in land surface models. By embracing high-resolution modeling, we can enhance our understanding of Earth's systems and their responses to climate change.

# 1   Introduction

The Earth's changing climate is causing increasingly severe impacts on ecosystems worldwide (Pachauri et al., 2014; IPCC, 2022). Human activity has played a significant role in past land-cover changes and will continue to have both direct and indirect impacts in the future (Vitousek et al., 1997; Pitman, 2003; Sterling et al., 2013; Pongratz et al., 2021). In the context of the climate system, land surface processes control the exchange of carbon, nitrogen, and energy between the atmosphere and the



terrestrial Earth, making them critical components of the current and future climate (Ferguson et al., 2012; Dirmeyer et al.,
    2006; Seneviratne et al., 2006).

       Important interactions and feedback mechanisms exist between energy, water, and nutrient cycles. In seasonally snow-
    covered areas, the snowpack creates numerous such feedbacks: it controls the energy balance by modulating the exchange of
    heat and moisture between the land surface and the atmosphere (Thackeray et al., 2019). It determines the partitioning of energy

fluxes, influencing the magnitudes of both sensible and latent heat fluxes (Male and Granger, 1981), which, in turn, regulate
    the transfer of energy and water vapor, shaping the local and regional climate patterns (Ban-Weiss et al., 2011). Moreover,
    the duration and extent of snow cover has direct implications for vegetation periods, which has the potential to impact gross
    primary production (GPP), a measure of vegetation's ability to convert solar energy into chemical energy (and carbon dioxide
    to organic matter) through photosynthesis (Slatyer et al., 2022). Thus, the presence or absence of snow cover directly influences

the availability of water and sunlight for plants, influencing the productivity and carbon cycling within terrestrial ecosystems.

       As snow plays a vital role in the Earth's energy balance (e.g., due to its high reflectivity (Flanner et al., 2011) and low ther-
    mal conductivity (Zhang, 2005)) and hydrological cycle (Flanner and Zender, 2005; Barnett et al., 2005), understanding and
    quantifying the intricate interactions among snow cover and ecophysiological processes is essential for accurate predictions of
    environmental change and its impacts on the Earth's systems. Experimental studies, including snow manipulation experiments

(Rixen et al., 2022; Slatyer et al., 2022), have observed and assessed these feedbacks at the local scale (e.g., Zeeman et al.
    (2017); Cooper et al. (2020)); Extrapolating these findings to regional and global scales, however, is only possible through
    modelling studies and remains challenging today. Ecohydrological models such as Tethys&Chloris (Fatichi et al., 2012; Mas-
    trotheodoros et al., 2020) and RHESSys (Son and Tague, 2019; Tague and Band, 2004) are specifically designed to represent
    interactions between water, energy, and the carbon cycles, but are not suitable for global-scale applications.

Land surface models (LSM), in contrast, specifically target global-scale applications, as they were initially developed to
    represent the lower atmospheric boundary condition of Global Circulation Models. Land surface modeling has seen remark-
    able progress in recent years, evolving from simple biophysical parametrizations to complex frameworks that incorporate key
    processes such as soil moisture dynamics, land surface heterogeneity, and plant and soil carbon cycling (Fisher and Koven,
    2020; Lawrence et al., 2019). Today's LSMs are thus principally suitable for, and even intended to, study process interactions

and feedbacks within the Earth's systems (e.g., Lawrence et al. (2019). However, large challenges in land surface modeling
    today remain due to uncertainties in process representation, unresolved sub-grid heterogeneity, and the projection of spatial
    and temporal dynamics of model parameters (Beven and Cloke, 2012; Fisher and Koven, 2020; Fisher et al., 2019; Blyth et al.,
    2021). It is these limitations that make it difficult to reconcile site-scale experimental data and LSM simulations, hampering
    their evaluation and further development. Multi-resolution modelling setups (including the point/site scale) overcome this very

limitation, as it allows to evaluate a spatially distributed LSM simulation over a large spatial extent, while at the same time
    certain aspects of the model (i.e. snow depth / snow cover duration) can be validated at the point scale using in-situ observations.

       Today, a strong push is evident towards higher resolution modeling, such as 1km simulations (Schär et al., 2020). While
    achieving this level of resolution globally over extended periods remains a challenge due to computational limitations, higher
    resolution allows for a more precise representation of land surface heterogeneity, which directly influences the representa-





tion of various key parameters and their associated processes (e.g., Ma and Wang (2022); Rimal et al. (2019); Zhang et al. (2017)). Because snow cover dynamics is strongly affected by topography and thus highly variable in space (e.g., Clark et al. (2011)), higher resolution enables a more detailed characterization of snow distribution, depth, and duration, capturing the spatial variability of snow cover across diverse landscapes (Lei et al., 2022; Magnusson et al., 2019; Essery, 2003). Improved representation of snow cover dynamics has the potential to enhance simulation of surface albedo, which affects the amount

of solar radiation reflected back into the atmosphere, and thus influences surface temperatures (Thackeray and Fletcher, 2016; Flanner et al., 2011). Further variables and processes such as sensible and latent heat fluxes (Singh et al., 2015), surface temperature and evapotranspiration rates, and GPP, are highly variable in space (Anav et al., 2015). Increasing spatial resolution in land surface models, therefore has the potential to enhance not only the simulation of snow cover dynamics, but also helps understand for which related ecophysiological processes higher spatial resolutions is paramount. Improved representation of

the intricate interactions within the Earth's systems makes LSMs a powerful tool to study these feedback processes across scales and advance our understanding of them.

In this study, we explore how model resolution, and the quality of meteorological and land surface datasets affect the representation of snow cover dynamics and dependent ecophysiological variables. Based on the ideas highlighted above, we formulate the following hypotheses:

**Hypothesis 1**: *With increasing spatial resolution and quality of meteorological input datasets, the representation of snow cover dynamics and its associated variables in CLM5 can achieve an accuracy comparable to that of a dedicated snow model.*

However, differences in snow cover development (especially on the grid scale) raise the question of whether corresponding changes in growing season length arising from differences in simulated snow-cover have a substantial impact on phenology, ecosystem functions, and the water budget.

**Hypothesis 2**: *Higher spatial resolution and increased level of detail in input datasets systematically affect the simulation of snow cover-dependent ecophysiological variables. We therefore predict that an increasing spatial resolution also improves the simulation of evapotranspiration, and gross primary production, leading to better estimates of carbon fluxes.*

To test these hypotheses, we implement a multi-resolution modelling framework using CLM5, a state-of-the-art LSM. This framework bridges the gap between point/site-scale and spatially distributed land surface modeling, thus allowing us to compare

process representation accuracy across a hierarchy of spatial scales, while preserving model architecture. This way, confounding effects due to differences in process parametrizations are eliminated, isolating and clarifying the effects of model resolution and input-data, and allowing us to assess the importance of an accurate representation of sub-grid variability within coarser resolution models.

We apply our framework to the spatial extent of Switzerland, including relevant watersheds of neighboring countries. This

region provides an ideal setting due to its diverse topography, encompassing both the Swiss Alps and the Swiss plateau. We test our hypotheses 1 and 2 by investigating relative differences between different CLM5 configurations with regards to the (a) snow dynamics, (b) terrestrial carbon cycle by focusing on heat fluxes and photosynthetic activity (GPP) and (c) the terrestrial water budget by focusing on the sum of water returning back to the atmosphere (evapotranspiration, ET). Our findings can inform



further offline applications of LSMs to 1) extrapolate local-scale experimental findings, and 2) provide context to global-scale, coarse resolution simulations.

## 2    Methodology

### 2.1    Land surface modelling

To investigate the effects of spatial resolution and input datasets in LSMs, we use the land component of the Community Earth System Model (CLM5), a state-of-the-art LSM that simulates carbon, nitrogen, water and energy exchange between the atmosphere and the land surface (Lawrence et al., 2019, 2018). It offers two operational modes: prognostic biogeochemistry (BGC) mode, which fully prognostically calculates all state variables, and prescribed satellite phenology (SP) mode. For this study, we focused on running CLM5 in SP mode, where latest remote sensing-based datasets are used to prescribe part of the state variables in natural vegetation, reducing the degrees of freedom compared to prognostic calculations. It's important to note that in SP mode, carbon-nitrogen cycling is not considered, and certain processes such as leaf nutrient limitation and respiration terms are omitted. GPP in CLM5 SP is approximated by photosynthetic activity, with photosynthesis being limited by carboxylation, light, and export limitations for different plant functional types (Thornton and Zimmermann, 2007; Farquhar et al., 1980). The photosynthesis module in CLM5 is described in detail by Thornton and Zimmermann (2007), Bonan et al. (2011), and Oleson et al. (2010). Evapotranspiration in CLM5 is calculated as the sum of transpiration, evaporation (considering soil/snow evaporation, soil/snow sublimation as well as dew), and canopy evaporation following Lawrence et al. (2007).

Snow cover provides a convenient means of observing and validating the internal energy turnover of LSMs, and it is the duration of snow cover that influences vegetation periods, ecophysiological processes, and carbon cycles. In CLM5, general snow parametrizations are based on Anderson (1976), Jordan (1991), and Dai and Zeng (1997), with fractional snow cover calculations being based on the method of Swenson and Lawrence (2012). In recent years there have been several updates to the snow-related parametrizations, most notably an inclusion of wind and temperature effects on fresh snow density and an increase in maximum snow layers from 5 to 12 (Lawrence et al., 2019). A detailed description of snow related calculations in CLM5 can be found in Lawrence et al. (2018).

Spatial resolution mostly influences the representation of spatial heterogeneity in CLM5 which is represented by a sub-grid hierarchical system. Each grid cell is split into different land units (vegetation, glacier, lake, urban, crop), and vegetated land-units further divided up into 15 Plant Functional Types (PFTs) or bare ground (this is the second sub-grid level in CLM5, often referred to as the column-level). Vegetation structure for each PFT is described by monthly varying Leaf Area Index (LAI) and Stem Area Index (SAI), as well as canopy top and bottom heights. Here, we applied CLM5 both to the regional scale, and to the point-scale, for which CLM5 features a dedicated point mode (PTCLM). It is worth noting that what we refer to as point-scale simulations incorporates fractional state variables (e.g., fractional snow-cover), as the gridded modeling algorithms (e.g., exactly the same algorithms that are applied to large-scale gridded simulations) are directly applied to a single point. From a snow-cover modeling perspective such an approach would be referred to as site-scale, but in order to be consistent with LSM conventions we refer to them as point-scale simulations.



## 2.2 Model experiments with CLM5

Figure 1 provides a general overview of the experimental setup, which includes three main aspects. Firstly, we varied the spatial resolution, ranging from 0.5°(10x6 grid cells) to 0.25°(19x11 grid cells) to 1 km (365x272 grid cells) over the study domain. Secondly, we used different meteorological forcing datasets, including a globally available coarse-resolution dataset (Clim$_{CRU}$), the same global dataset with lapse-rate corrected temperature (Clim$_{CRU*}$), and a high-resolution regional dataset (Clim$_{OSHD}$). Lastly, we considered two options for land-use information: a global dataset (LU$_{Gl}$) and a high-resolution dataset (LU$_{HR}$). This approach is intended to cover multiple facets of resolution: on one hand, the spatial resolution of the CLM5 simulations themselves; on the other hand, the 'native' resolution, or level of detail, of the input datasets, with higher resolution implying better quality of the datasets. Different CLM5 configurations were set up to cover the variations in spatial resolution, meteorological forcing, and land-use information.

At the 1km scale, CLM5 was run with six different configurations, each using different combinations of meteorological forcing and land-use information. At the 0.5°and 0.25°resolutions, CLM5 was run with three configurations corresponding to the respective meteorological forcing datasets and using the global land-use dataset. These regional CLM5 simulations across the spatial extent of Switzerland and adjacent watersheds of neighboring countries, covering an area of 44,050 km$^2$, were set up in an identical way as global simulations.

Additionally, point-scale simulations were conducted at 36 snow-monitoring station locations within the model domain. At the snow monitoring stations, we focus on the impact of meteorological forcing and land-surface input on CLM5 simulations by running the same six configurations as for the 1km gridded experiment. Land surface information for each site location was thereby extracted from the nearest 1km tile of the gridded dataset (either LU$_{Gl1km}$ or LU$_{HR1km}$). Model performance evaluation was carried out based on in-situ observations at these stations (see Section 2.4.1 and 2.5.1 for more information). We also set up simulations at 6 FLUXNET tower locations (Pastorello et al., 2020), setup and results of which can be found in Appendix B.

The performance of all gridded CLM5 configurations in simulating seasonal snow cover was assessed against simulations obtained with a the dedicated snow model (see Section 2.4.2 and 2.5.2 for more information). Outcomes from the snow cover analyses were complemented by examining the link between spatially distributed CLM5 simulations of seasonal snow and their subsequent effects on ecophysiological variables through a relative comparison of the different gridded CLM5 model configurations, with a particular focus on gross primary production and evapotranspiration.

## 2.3 Input datasets

Each CLM5 model configuration requires the following meteorological driving data: incident short and long-wave radiation, air temperature, relative humidity, wind speed, pressure, and precipitation. Additionally, a land surface information file is required.

CLM5 simulations were set-up to run between January 2016 and December 2019, in order to maximize the temporal overlap between the various meteorological forcing datasets and available data for model bench-marking. We further performed 10 years of spin-up in accelerated decomposition mode, followed by a final spin-up of 10 years, both by re-cycling through the

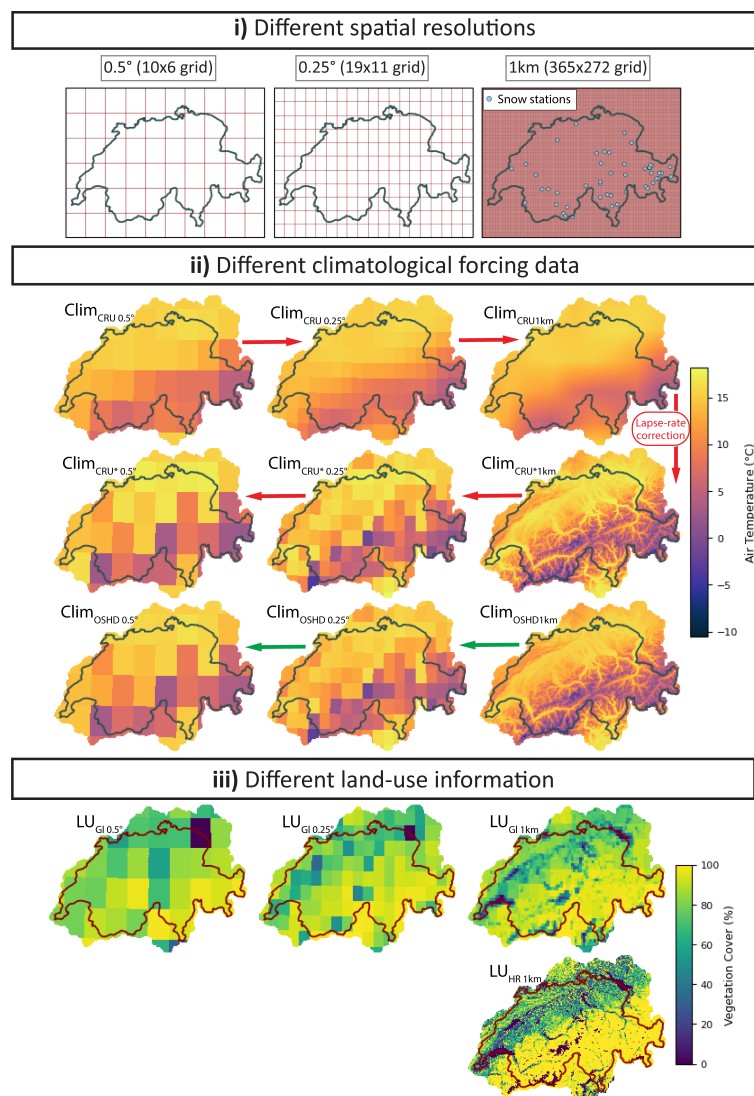

**Figure 1.** Schematic overview specifying the 3 facets of the experimental setup: Variation of i) spatial resolution, ii) meteorological forcing data and iii) land-use information. i) shows the different grids used, including the locations of the snow stations. ii) shows monthly mean temperature (May 2018) from the different data sources: Globally-available coarse-scale dataset (Clim$_{CRU}$), the same but with a lapse-rate corrected temperature (Clim$_{CRU*}$), and a high-resolution regional dataset (Clim$_{OSHD}$). Note that Clim$_{CRU}$ data is provided at 0.5°(top left-most panel in ii), and bilinearly regridded to 0.25°and 1km. Clim$_{CRU1km}$ is then downscaled via a lapse-rate correction to obtain Clim$_{CRU*1km}$, before being up-scaled to 0.25°and 0.5°. Apart from temperature, meteorological forcing data is identical for Clim$_{CRU1km}$ and Clim$_{CRU*1km}$ simulations. Clim$_{OSHD}$ data is provided at 1km, and upscaled to 0.25°and 0.5°. iii) shows differences in land-use information considered in this study by the example of percentage vegetation cover.





available input-data. A spin-up was necessary to ensure soil moisture and soil temperature were in approximate equilibrium
and not affecting temporal dynamics and physical properties e.g., of the simulated snow cover evolution.

### 2.3.1 Meteorological Forcing

To assess the impact of meteorological input data quality, we considered three meteorological forcing datasets with increasing
level of detail. As an example of a standard global dataset, we used the recent state-of-the-art dataset CRU-JRA (University

of East Anglia Climatic Research Unit; of East Anglia Climatic Research Unit; Harris (2019)), which provides near-global
(excluding Antarctica) six-hourly meteorological data on a 0.5°latitude x 0.5°longitude grid. CRU-JRA is a merged product
of the monthly Climate Research Unit (CRU) gridded climatology (Harris et al., 2014) with the Japanese Reanalysis product
(JRA, Kobayashi et al. (2015)). We selected CRU-JRA due to its large timespan (1901-2020), which includes recent years and
hence ensures sufficient overlap with our high-resolution forcing dataset (see below), data as well as due to its application in

the annual Global Carbon Budget assessments (e.g., TRENDY, Friedlingstein et al. (2020)) in the Land Surface, Snow and Soil
Moisture Model Intercomparison Project (LS3MIP, Hurk et al. (2016)). The original 0.5°CRU-JRA dataset was first projected
to our model domain using nearest neighbor techniques ($\text{Clim}_{\text{CRU0.5°}}$), before re-gridding it to 0.25°and 1km using bilinear
interpolation to obtain $\text{Clim}_{\text{CRU0.25°}}$ and $\text{Clim}_{\text{CRU1km}}$.

As a dataset representing an intermediate level of detail, we upgraded the $\text{Clim}_{\text{CRU1km}}$ dataset by downscaling temperature

data using a temperature lapse rate of -6.5K/1000m, which resulted in the $\text{Clim}_{\text{CRU*}}$ dataset. This approach was intended to
account for variations of air temperature within the complex topography of the Swiss Alps and subsequent refinement of the
partitioning of precipitation into snow and rain. The updated 1km fields were upscaled back to 0.25°and 0.5°to inherit this
correction also to the coarser-resolution simulations. This resulted in the $\text{Clim}_{\text{CRU*}}$ dataset. All other forcing variables were left
identical for $\text{Clim}_{\text{CRU1km}}$ and $\text{Clim}_{\text{CRU*1km}}$ simulations.

As input datasets with the highest level of detail, we used meteorological forcing generated according to methods developed
by the Operational Snow Hydrological Service (OSHD), at 1km spatial and 1hour temporal resolution as well as all point
locations at 1 hour temporal resolution. Necessary meteorological input variables were all provided by MeteoSwiss (COSMO1
and COSMOE product), and specific downscaling routines were applied e.g., to incoming solar radiation and wind velocity
to optimally capture the influence of complex topography. Of particular relevance to this study is the correction of snowfall

input fields by assimilation of station data according to Magnusson et al. (2014). In the context of this study, this dataset can be
considered a meteorological input specifically optimized for accurate gridded snow cover simulations. The 1km forcing data
was then upscaled to the desired target resolution (0.25°and 0.5°) with no smoothing applied. We refer to Mott et al. (2023)
for further details with regards to the $\text{Clim}_{\text{OSHD}}$ product.

### 2.3.2 Land-use information

Input datasets for the land surface are based on the global-scale input dataset commonly used in CLM5, where percent plant
functional type for each grid cell are derived from MODIS satellite data (Lawrence and Chase, 2007), as are monthly LAI
and SAI values. These global-scale surface input-datasets have an initial resolution of 0.05°. We firstly re-projected and re-



gridded the dataset to the model domain using Earth System Modelling Framework (ESMF) regridding tools, which resulted in our "global info" land surface dataset (LU$_{Gl}$, see Figure 1) and represents a land surface dataset equivalent to that which

would be used in a typical large scale LSM/General Circulation Model application. This step was performed separately for each target resolution, resulting in the LU$_{Gl0.5°}$, LU$_{Gl0.25°}$ as well as the LU$_{Gl1km}$ dataset. To obtain an alternative land-use input dataset with a higher level of detail, the LU$_{Gl1km}$ dataset was updated based on a combination of the official land use and land cover Data of Switzerland (Arealstatistik, 100m resolution, updated every 6-8 years and derived by visual interpretation of aerial photographs (Swiss Federal Statistical Office, 2001), forest mixing ratios (100m resolution, Swiss Federal Statistical

Office (2013)) and Copernicus Sentinel-3 data (333m resolution). More specifically we merged arealstatistics Switzerland and Copernicus Sentinel-3/OLCI, PROBA-V @333m and forest mixing ratios to obtain vegetation, lake, urban, glacier, crop-fraction at the land-unit level and monthly LAI, SAI, fraction per PFT (incl. bare ground) at the column level. This resulted in the high-resolution LU$_{HR1km}$ dataset.

## 2.4 Test datasets

We used two observational datasets as test datasets to assess model performance. The first, consisting of daily snow depth observations from 36 snow stations, served to evaluate the performance of CLM5 point-scale configurations in simulating seasonal snow cover. For the second dataset, we employed the Flexible Snow Model (FSM2) as a reference snow model for validation and comparison with the gridded CLM5 simulations of seasonal snow development.

### 2.4.1 Snow stations

The 36 snow stations considered cover an elevational gradient and are spread throughout Switzerland (see Figure 1i)). Table A1 in the Appendix specifies locations and characteristics of each of these sites. Observations at the station locations consist of daily monitored snow depth (HS), which are collected as part of the snow monitoring networks of either the WSL Institute for Snow and Avalanche Research (SLF) or the Federal Office for Meteorology and Climatology (MeteoSwiss). HS measurements were extracted at a daily timestep and cleaned from obvious outliers (assessed against neighboring stations at similar

elevations), which can occur e.g. due to transmission or measurement errors (see Mott et al. (2023)).

### 2.4.2 Snow cover simulations with FSM2

The Flexible Snow Model (FSM2, Mazzotti et al. (2020)), a recent upgrade of the Factorial Snow Model (FSM, Essery (2015)), is an open-source, spatially distributed, physics-based snow model. Gridded simulations at 250m resolution and point simulations at snow station locations have been specifically set up and calibrated by SLF to run over the extent of Switzerland for

the purpose of operational snow water resources monitoring (Griessinger et al., 2019; Mott et al., 2023). In the absence of high-quality, spatially distributed snow depth observations over the entire extent of Switzerland, these FSM2 simulations were served as ground truth for this study. For comparison with CLM5 output, 250m resolution FSM output results were upscaled to 1km without smoothing.





### 2.5 Evaluation of model performance

#### 2.5.1 Comparing point-scale CLM5 model simulations to station observations of snow depth

Observations at the snow monitoring stations (Figure 1i and Table S1) were used to assess the ability of each point-scale CLM5 configuration to simulate seasonal snowpack in Switzerland and were additionally compared to offline FSM2 simulations. The stations were binned into three elevational bands (<1000 m.a.s.l, 1000 – 2000 m.a.s.l, >2000 m.a.s.l) resulting in 10, 12 and 14 stations for the low, mid- and high elevation band, respectively. For each station location, the various CLM5 point-scale

simulations ($Clim_{CRU1km}+LU_{Gl/HR\ 1km}$, $Clim_{CRU*1km}+LU_{Gl/HR\ 1km}$, $Clim_{OSHD1km}+LU_{Gl/HR\ 1km}$) as well as the FSM2 simulation were compared to observations of snow depth (HS), by computing relative and absolute differences as well as Root Mean Square Errors (RMSE) and Mean Absolute Errors (MAE) for the timeframe between November and May of each simulation year (2016-2019).

#### 2.5.2 Comparing gridded CLM5 model simulations to FSM2 simulations of snow depth

For the evaluation and quantification of snow-related CLM5 model experiment's performance we used a Taylor diagram (Taylor, 2001), with FSM2 simulations of snow depth as our reference. A Taylor diagram combines centered RMSE, correlation coefficients as well as the spatial/temporal standard deviation and hence describes overestimation or underestimation of the models relative to a benchmark. In order to calculate the values for the Taylor diagram the output of the low resolution CLM5 simulations (0.25° and 0.5° resolution) was interpolated to the finer 1km grid without smoothing.

## 3 Results

### 3.1 Snow simulations

#### 3.1.1 Snow dynamics using different meteorological forcing

We observed distinct differences in performance using different meteorological forcing datasets in our CLM5 experiments (see Figure 2). The CLM5 model using global meteorological forcing data ($Clim_{CRU1km}+LU_{Gl/HR\ 1km}$) showed poor performance in

modeling seasonal snow development. RMSEs exceeded 1m for mid-elevation stations and only marginally improved for high- and low-elevation stations. When the lapse-rate based downscaled temperature input was used ($Clim_{CRU*1km}+LU_{Gl/HR\ 1km}$) instead, the model's performance improved significantly, particularly at low elevations. The CLM5 model forced with OSHD data ($Clim_{OSHD1km}+LU_{Gl/HR\ 1km}$) demonstrated the best performance across all three elevation bands, with only minor errors in low- and mid-elevation locations (e.g., RMSE/MAE of 0.25/0.15m for mid-elevation $Clim_{OSHD1km}+LU_{HR\ 1km}$ simulations).

Results were consistent throughout all simulated years.



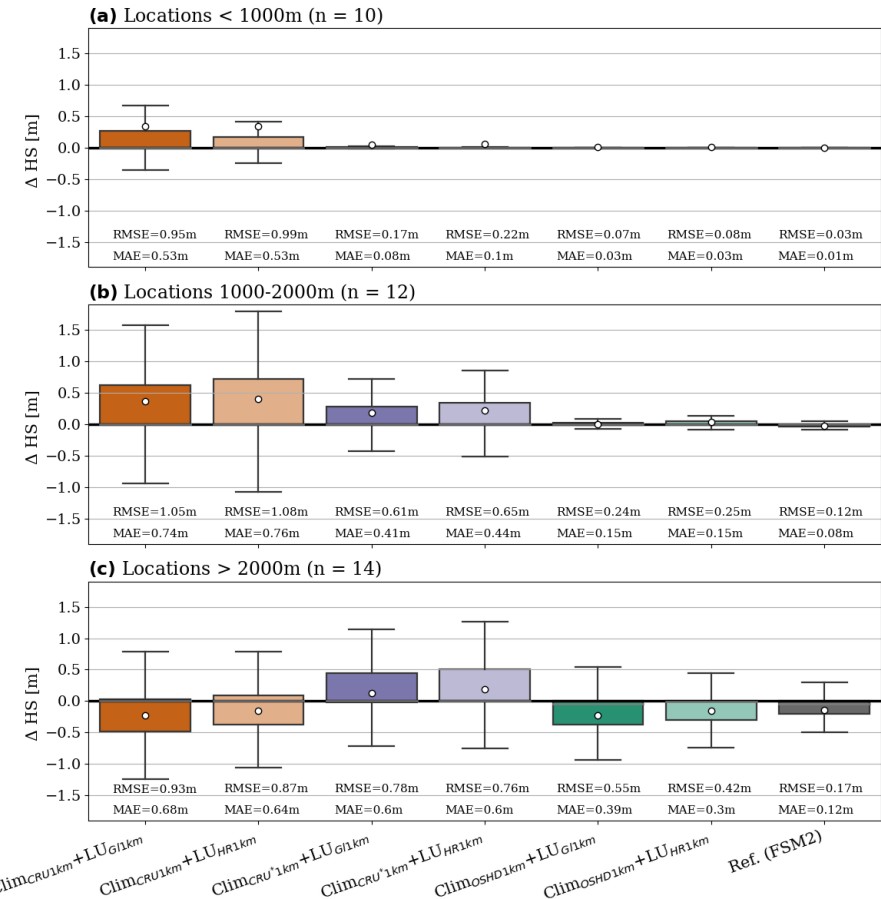

**Figure 2.** Comparisons of point-scale model simulations to observations of snow depth (HS) during the 2017/18 snow season (November-May) for combined (a) low elevation, (b) mid-elevation and (c) high elevation snow station locations. Negative values depict underestimations of the simulations. Means are shown by the white dots. The reference snow simulation (FSM2) matches observations the closest, with negligible errors for low and mid-elevation points and slight underestimation for high elevation points. CLM5 forced with OSHD data and based on high-resolution land-use information is the next best. For a more detailed assessment of seasonal snow dynamics per station and simulation, refer to Figure A1 in the Appendix.

### 3.1.2 Snow dynamics using different land use information

Regarding the effects of the land-use information dataset, we observed that the choice of land-use information had a smaller impact compared to the meteorological forcing data (Figure 2). While a slight improvement was seen when using the high-resolution land-use information dataset ($LU_{HR1km}$) at high elevations for all three sets of meteorological forcing data (reducing RMSE by -0.13m/-0.02m/-0.06m for $Clim_{CRU1km}$/ $Clim_{CRU*1km}$/ $Clim_{OSHD1km}$ simulations, respectively), no significant differ-



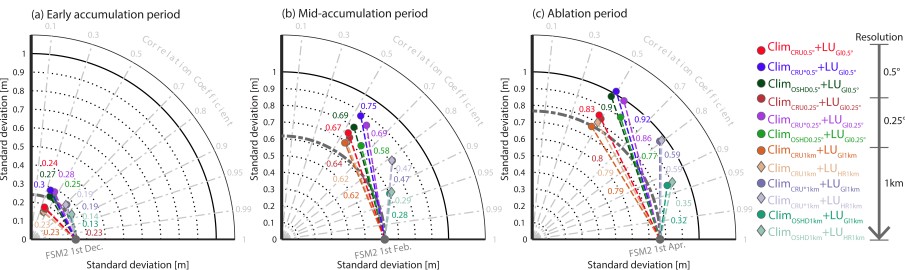

**Figure 3.** Taylor plots (Taylor, 2001) for comparisons of simulated snow-depth (HS) between all 12 different CLM5 configurations and the reference snow simulation (FSM2, dark grey) during (a) early accumulation season (1-Dec), (b) mid-accumulation period (1-Feb) and (c) ablation period (1-Apr) throughout four winter seasons (2015/16, 2016/17, 2017/18, 2018/19). The plotted statistical metrics allow for evaluation and quantification of CLM5 model experiments performance based on centered RMSE (directly proportional to the distance away from the reference (=FSM2)), correlation coefficients (azimuthal position) and the spatial/temporal standard deviation (radial position from the origin) which determines overestimation or underestimation of the models. An increase in resolution results in improved representation of snow for all 3 time periods and across all meteorological/land-use information combinations. Simulations with high-resolution meteorological forcing data substantially outperform global meteorological forcing cases, whereas simulations with different land-use information only differ marginally. $Clim_{OSHD1km}+LU_{Gl1km}$ performs closest to the reference FSM2 simulation during all 3 time periods. The global case simulations (e.g., $Clim_{CRU0.5°}+LU_{Gl0.5°}$) do not perform well with low correlations and high standard deviations. See Figure 1(ii) and (iii) for the various combination setups.

ences or marginal decreases in model performance were observed for $Clim_{OSHD1km}$, $Clim_{CRU1km}$ and $Clim_{CRU*1km}$ across all elevation bands.

### 3.1.3 Accuracy of FSM2 point-scale simulations

Across all elevation bands, the FSM2 simulations closely matched the observations, with only minor errors at low and mid
elevations during the 2017/18 season (Figure 2). At high elevations, the FSM2 model slightly underestimated snow depths, which can be assessed in more detail in Figure S1 in the supplementary material. Figure A1 in the Appendix features wiggle plots as well as absolute difference and seasonal snow development for selected snow station locations, visualizing the superior performance of FSM2 in comparison to various CLM5 model experiments, further justifying using FSM2 model simulations as our ground truth for the gridded simulation comparisons in Section 3.1.4.

### 3.1.4 Spatially distributed snow cover

Here we investigate all 3 facets of this study: Effects of resolution, effects of meteorological forcing data, and effects of land-use information data. We focus on gridded simulations of snow depth from all 12 different CLM5 configurations (see Figure 1(ii) and (iii)) and compare them to FSM2 simulations. Increasing the level of detail in meteorological forcing data plays the largest role in modulating accuracy of simulated seasonal snow cover. CLM5 runs with OSHD-based input data outperform all
CRU- and CRU*-based simulations at all three points in time during winter (e.g., RMSE $Clim_{OSHD1km}+LU_{Gl1km}$: 0.13, 0.28,





0.32m vs. RMSE $\text{Clim}_{\text{CRU*1km}}+\text{LU}_{\text{Gl1km}}$: 0.19, 0.47, 0.59m vs. RMSE $\text{Clim}_{\text{CRU1km}}+\text{LU}_{\text{Gl1km}}$ : 0.23, 0.62, 0.79m for early, mid, and end-winter respectively).

When running CLM5 with global-based forcing data, increasing spatial resolution in isolation (e.g., regridding) only has a marginal effect on accuracy of simulated seasonal snow cover during early and mid-winter, with a bit more of a pro-
nounced effect of increases to 1km during the ablation period (see difference between $\text{Clim}_{\text{CRU0.5°}}+\text{LU}_{\text{Gl0.5°}}$ (red dots), $\text{Clim}_{\text{CRU0.25°}}+\text{LU}_{\text{Gl0.25°}}$ (dark red dots) and $\text{Clim}_{\text{CRU1km}}+\text{LU}_{\text{Gl1km}}$ (orange dots) in Figure 3 a,b,c). The marginal effect can be attributed to the fact that increasing spatial resolution in itself (e.g., simple regridding) does not bring any added value as in better representation of topography. However, when using the down-scaled global temperature data as well as the OSHD forcing data, there is a substantial reduction in accuracy between the 1km and the 0.5°/0.25° simulations (Figure 3), implying
that a coarse resolution negates the benefit of a higher level of detail in the meteorological forcing.

Similarly to the point-scale simulations at the snow station locations, the choice of land-use information only had a marginal influence on the simulation accuracy of seasonal snow-cover development. Ultimately, throughout the 4 modeled years, and averaged over the model domain, substantial differences in simulated snow-cover between the various CLM5 configurations are prevalent (Figure 3). In a similar manner to the point-scale CLM5 simulations, results revealed vast improvements in simulated
snow cover accuracy when using high-confidence forcing data (Figure 2, Figure 3), with CLM5 in our best-effort scenario ($\text{Clim}_{\text{OSHD1km}}+\text{LU}_{\text{HR1km}}$ simulation) almost reaching the level of a dedicated snow model. This becomes especially apparent when looking at the high correlation coefficient of the $\text{Clim}_{\text{OSHD1km}}+\text{LU}_{\text{HR1km}}$ simulation in Figure 3.

### 3.2 Simulation of ecophysiological parameters

A relative comparison between spatially distributed (a) simulated mean total peak (July+August) growing season GPP for
2017 as well as (b) total ET during 2017 is shown in Figure 4. In each plot $\text{Clim}_{\text{OSHD1km}}+\text{LU}_{\text{Gl1km}}$ is compared with the $\text{Clim}_{\text{OSHD1km}}+\text{LU}_{\text{HR1km}}$ simulation (effect of land-use information), with the $\text{Clim}_{\text{CRU1km}}+\text{LU}_{\text{Gl1km}}$ (effect of meteorological forcing) as well as with the $\text{Clim}_{\text{OSHD0.5°}}+\text{LU}_{\text{Gl0.5°}}$ simulation (effect of spatial resolution). Figures C1-C4 in the Appendix show this relative comparison for all remaining CLM5 model configurations used in this study, while Figure C5 shows monthly values for the full simulation period, averaged over the model domain. For GPP, effects of land-use information are pronounced,
with a mix of over- underestimations across the model domain. Meteorological forcing data had a slightly smaller with relative overestimations of GPP throughout the model domain (up to 14% during peak growing season when averaged over the model domain, see Figure C5). For the coarse-scale runs, we see that non-resolved surface heterogeneity (e.g., lakes) has a large effect on simulated GPP, underlying the effect of resolution.

The choice of land surface information datasets, on the other hand, only showed marginal effects on simulated ET, but the
effect of meteorological forcing results in substantial differences in simulated ET (up to 26% when averaged over the entire model domain, see Figure C5 in the Appendix). This effect is especially pronounced along the Swiss Alps, where complex terrain leads to differences in precipitation patterns captured by the two forcings (see Figure C6 in the Appendix for a direct comparison of precipitation patterns in the forcing datasets). Similarly to GPP, the effect of resolution in isolation strongly



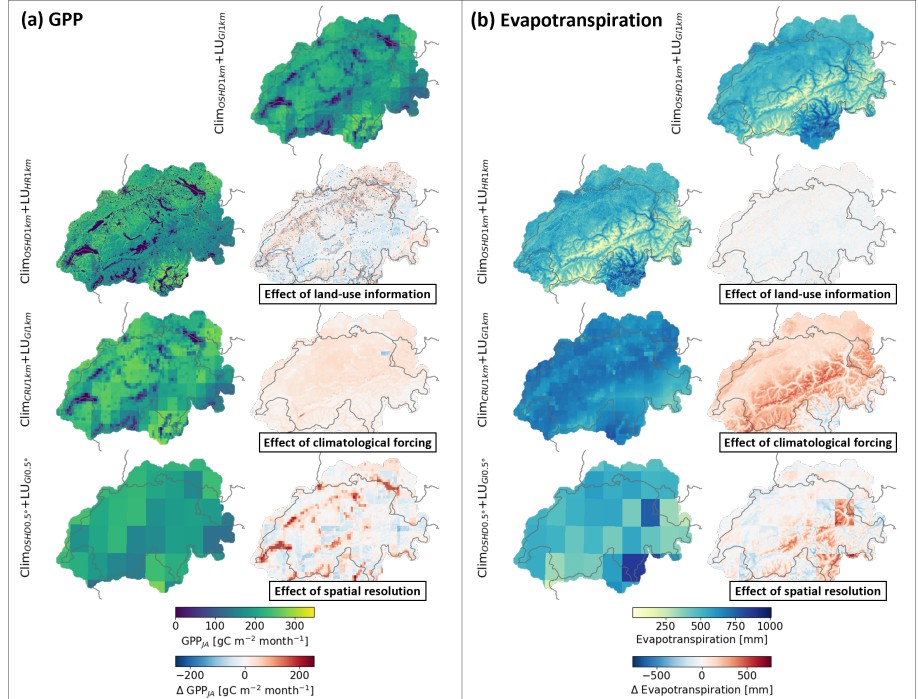

**Figure 4.** Spatial comparison of CLM5-simulated a) GPP and b) Evapotranspiration for the following cases: i) denotes the reference case, with "best"climate (Clim$_{OSHD1km}$) + global land-use info (LU$_{HR1km}$), ii) singles out the effect of land-use information and iii) shows the combined effect climate and land-use info. For the residual plots, blue indicates underestimation and red indicates overestimation with regards to the reference case.

affects the patterns of simulated ET due to non-resolved surface heterogeneity, but is less directional and hence difficult to
quantify

### 3.3   Seasonal snow cover development and ecophysiological variables

Substantial differences in simulated snow-cover as well as in simulated ecophysiological variables persist between the various CLM5 configurations (see Section 3.1 and 3.2, respectively). These demonstrated differences raise the question of the link between these discrepancies, more specifically whether corresponding changes in growing season length arising from differences
in simulated snow-cover have substantial impacts on the simulated terrestrial carbon cycle and water budget. Here, we focus on differences between the best-effort simulation (Clim$_{OSHD1km}$+LU$_{HR1km}$) and the CLM5 configuration with global meteorological forcing data (Clim$_{CRU1km}$+LU$_{HR1km}$, 'effect of meteorological forcing data') where large differences in the snow-based evaluation were evident (see Section 3.1), asking whether these differences are correlated with simulation differences in gross primary production and evapotranspiration.





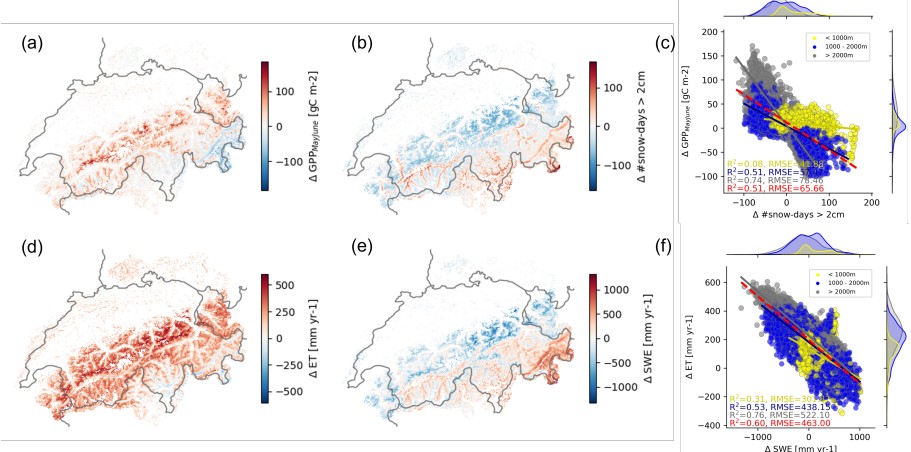

**Figure 5.** a-b and c-d: Spatial comparison of differences in CLM5 simulations highlighting the cascading effects of changes in snow cover development to changes in ecophysiological variables. We compare our best-effort reference case (Clim$_{OSHD1km}$+LU$_{HR1km}$) and Clim$_{CRU1km}$+LU$_{HR1km}$ for 2017 showing a) monthly-averaged GPP in May and June 2017, b) number of days with more than 2cm of snow on the ground between January and July 2017, d) yearly sum of Evapotranspiration between October 2016 and September 2017, and e) yearly cumulative sum of Snow Water Equivalent (SWE) between October 2016 and September 2017 (total positive SWE increments; 'how much water is stored in total'). c) and f) show the correlation between GPP and number of snow days and ET and total SWE for 3 elevation bands (<1000m = yellow, 1000-2000m = blue, >2000m = grey) including regression lines. The dashed red line in c) and f) is the overall regression line. For the residual (scatter) plots, blue (negative) indicates underestimation and red (positive) indicates overestimation with regards to the reference case.

Simulation differences in monthly-averaged GPP during May and June are shown to be negatively correlated with simulation differences in snow duration (number of days with more than 2cm of snow on the ground between January and June, Figure 5a-c), which becomes apparent when visually comparing spatially explicit differences (Figure 5a-b): It is evident that locations with lower differences in GPP production coincide with a larger difference in snow duration and vice-versa. This relationship is strong for elevations above 2000m (R2 = 0.74) and relatively strong between 1000 and 2000m (R2 = 0.51). Expectedly, for pixels below 1000m this relationship is less pronounced, which can be attributed to the reduced effect of snow-cover in low elevations. More generally we confirm the hypothesis that seasonal snow cover is important for determining productivity during the growing season length, with a negative correlation between snow and GPP (see Figure C7).

Shifting from the terrestrial carbon cycle to the water budget, we focus on simulation differences in yearly evapotranspiration (ET) and the total yearly amount of water contained within the snow-pack (snow water equivalent, SWE) to investigate whether differences in ET can be explained by snow on the ground in addition to differences in precipitation input itself. Evapotranspiration differences are shown to be negatively correlated with total SWE differences across all elevation bands (Figure 5d-f), underlining the importance of snow for quantifying water feedbacks to the atmosphere with CLM5.



A summary statistic of the four discussed variables in this section (GPP$_{MayJune}$, number of snow days, total ET, total SWE) for all CLM5 configurations of this study is provided in Figure C8 (Appendix), which underlines that our best-effort reference case (Clim$_{OSHD1km}$+LU$_{HR1km}$) exhibits a wider spread in simulated GPP$_{MayJune}$ and total ET when compared to remaining CLM5 configuration simulations, which reflects the wider spread of snow-duration. Figure C9 additionally shows summary statistics for GPP$_{JulyAugust}$, which when compared to Figure C8c) (GPP$_{MayJune}$) suggests that snow cover impacts on GPP are mainly visible in the beginning of the growing season, further explaining why GPP$_{MayJune}$ was used here.

## 4 Discussion

In this study, we used a multi-resolution modeling setup to examine how input data and spatial resolution affect the accuracy of seasonal snow cover simulations and their impact on ecophysiologial variables, with the focus being gross primary production (GPP) and evapotranspiration (ET). Availability of a wealth of snow station data in combination with operational FSM2 results provided a unique opportunity to systematically assess snow cover simulation accuracy across elevations, in a spatially and temporally explicit manner. We found that the most accurate snow cover simulations for Switzerland, with results comparable to those of the operational snow-hydrological model, were achieved using high-resolution meteorological forcing data (OSHD) and a 1km resolution that fully resolved landscape heterogeneity confirming our hypothesis 1, which aligns with previous studies (e.g., Lüthi et al. (2019)). Simulation performance of snow-cover simulations were dictated by the capability of the meteorological input to capture topographic effects (e.g., improved estimation of precipitation phase due to the high resolution temperature fields) and precipitation patterns, which is a function of both input type (e.g., OSHD vs. CRU) and model resolution (aggregating OSHD-based forcing data for coarser resolution simulations drastically reduced simulation accuracy).

We observed a negative correlation between differences in growing season length (quantified as number of snow-free days) and GPP estimates across the model domain, highlighting the significance of accurate snow cover simulations for the carbon budget, and confirming hypothesis 2. The link between snow duration and GPP was, however, much stronger during the early growing season (May-June), as compared to peak growing season (July-August), suggesting that additional differences between the CLM5 model configurations that happen in the summer period confound the effect of snow alone. Such differences can arise e.g. due to discrepancies in summer precipitation captured by the different meteorological forcings, or because inconsistencies in vegetation parameters derived from different land use datasets result in different magnitudes of ecophysiologial processes.

The variations in estimated peak summer GPP between different CLM5 configurations exceeded 200 gC m-2 month-1, equivalent to a 8,810 tC month-1 difference in peak growing season GPP across Switzerland. These differences are noteworthy because errors in simulated GPP can propagate through LSMs and introduce additional errors in simulated biomass and other fluxes (Schaefer et al., 2012). They have substantial implications for climate predictions and underline the challenges that current LSMs face in predicting carbon exchanges (Beer et al., 2010; Zaehle et al., 2014). Our findings regarding GPP modeling uncertainties are consistent with a recent study that showed CLM5 overestimated summer GPP by up to 40% in an arctic boreal environment compared to observational data (Birch et al., 2021). This study emphasizes the importance of regional model analysis and development.





Another crucial variable in linking global water, carbon, and energy cycles in LSMs is ET. ET is important for the water budget, while at the same time being a relevant process for energy balance (e.g., latent heat flux). We detected large effects of spatial resolution and choice of input data on ET estimates over the model domain, with the choice of meteorological forcing data having the largest effect. Compared to the mountainous, high-elevation areas of Switzerland, increased ET was shown to

occur in the low-lands, in line with other studies which showed increased evapotranspiration as temperature increases (Zhao and Dai, 2015; Cheng et al., 2017). As ET is further limited by the amount of available moisture in the soil, and hence related to precipitation input as well as to seasonal snow, we compared differences in ET to differences in precipitation input itself, as well as to differences in total snow water. We were able to link differences in simulated ET between model configurations with regards to meteorological forcing data to differences in total snow water, highlighting the importance of rain-snow transitions

for ET calculations and further underlining results from Kraft and McNamara (2022). We were, however, not able to directly link differences between model configurations to differences in precipitation input. Significant positive correlation between precipitation and ET has been shown to predominantly occur in dry climates, where ET processes are controlled by water availability in the root zone or shallow surface (Shi et al., 2013), which is not the case in Switzerland. While some previous studies with CLM5 have shown that spatial distribution and interannual variability of ET is captured well in CLM5, when

compared to observations (Shi et al., 2013), and despite considerable progress in modelling terrestrial evapotranspiration in recent years, large uncertainties still exist (e.g., Miralles et al. (2016); Mueller et al. (2011)). The large differences we find in ET estimates across CLM5 configurations reflect this fact.

The finding that high resolution simulations improve the accuracy of seasonal snow cover development should thus not be interpreted as the main conclusion of our study. Instead, we have shown how biases in snow cover development can propagate

through the model and affect ecophysiological calculations. These findings are underlining local-scale results of e.g., Harpold (2016), which demonstrate the importance of snow-disappearance date for water stress across 62 sites in the Western US. They are further consistent with observations of greening as a result of reduced snow cover in Arctic (Myers-Smith et al., 2020) and Alpine (Rumpf et al., 2022) regions. At the same time, there is evidence that longer growing season is not always associated with increased productivity (Phoenix and Bjerke, 2016), highlighting the relevance of processes beyond the snow

season diagnosed in our analysis as well. Our study thus suggests the potential of LSMs applied at high spatial resolutions and fed with accurate input datasets to complement observational studies, as they allow us to quantify and better understand these still poorly constrained process dependencies across larger spatial and temporal extents, including predictions of future conditions.

To gain a more comprehensive understanding, it would be beneficial to repeat such a model experiment in an arctic en-

vironment rather than just an alpine one, as high latitudes are critical components of the rapidly changing climate system. Additionally, it is important to note that all simulations in this work were conducted in satellite phenology mode. Future studies should also compare CLM5 simulations with prognostic vegetation and biogeochemistry modes turned on to enable a more detailed analysis of the terrestrial carbon and nitrogen cycles, as well as evapotranspiration fluxes.

Uncertainty remains in LSM projections of climate change (e.g., Shrestha et al. (2022); Yuan et al. (2021, 2022)), with two

major sources of uncertainty being the effects of resolution and the quality of meteorological input data (especially precipita-



tion, Peters-Lidard et al. (2008)) on LSM simulation outputs. Quantifying such uncertainties is imperative to further increase the predictive power of climate impact models. Furthermore, given the complexity of state-of-the art LSMs, an understanding of the ways different parts/modules of LSMs interact with each other is more important than ever, as climate change impacts are not isolated, but highly interconnected processes (Zscheischler et al., 2018; Ridder et al., 2021). It is therefore of great importance to investigate how exchanges and interactions between model components are represented, rather than assessing process representation for each model component separately (Blyth et al., 2021), which ultimately requires multidisciplinary community efforts (Ciscar et al., 2019). Multi-resolution modelling frameworks as used for this study have large potential to help with such endeavors and provide critical insights into ecosystem responses to environmental change. More specifically, it can help identify both the key processes for which high spatial resolution and high-fidelity input data are necessary, as well as quantify the minimum resolution needed to resolve these processes accurately. Such modelling experiments should be prioritized in the future, ideally in combination with experimental manipulations (e.g., increase the availability of nitrogen or carbon dioxide in the system) as suggested by Wieder et al. (2019).

## 5 Conclusions

Using multi-resolution modeling experiments to quantify and potentially constrain uncertainties in land surface modeling, we highlight the importance of input data quality and spatial resolution in accurately representing processes across scales. By using regionally optimized datasets, we enhance the accuracy and applicability of LSM simulations, enabling a more comprehensive understanding of ecosystem responses to environmental changes. We could demonstrate the accuracy of simulated snow-cover in CLM5 simulations based on high-quality/high resolution meteorological forcing data and with landscape heterogeneity fully resolved at 1km and show how performance differences between different CLM5 configurations propagate through the model to result in substantial differences in gross primary production as well as evapotranspiration. The results clearly demonstrate the use of high spatial resolution and regionally detailed forcings in land surface models to better quantify and constrain the uncertainties in the represented processes, with profound implications for climate impact studies. More generally, this study highlights the utility of multi-resolution modeling experiments which bridge the gap between point-scale and spatially distributed land surface modeling when aiming to evaluate and improve process-based representation of variables in land surface models. Comparing process representation accuracy across a hierarchy of spatial scales, while preserving model architecture is therefore recommended for future land surface model developments.

*Code and data availability.* All scripts used for simulation setup and analysis can be found at https://github.com/johanna-malle/CLM5_CH. FSM2 snow simulation results can be downloaded from https://www.envidat.ch/dataset/seasonal-snow-data-wy-2016-2022. Upon publication, all CLM5 simulation results presented in this study will be available from the WSL data repository Envidat at their website under https://www.envidat.ch/.



## Appendix A: Point-scale CLM5 model simulations at snow stations

| Site | Name | Latitude (CH1903) | Longitude (CH1903) | Elevation [m a.s.l.] |
|------|------|-------------------|--------------------|-----------------------|
| BSG | Brissago | 108390 | 698200 | 280 |
| FRI | Frick | 262700 | 643353 | 345 |
| ALT | Altdorf | 191700 | 690960 | 449 |
| CBS | Chaebles | 186320 | 552495 | 589 |
| ABG | Labergement | 178770 | 527540 | 645 |
| MAS | Marsens | 167220 | 571440 | 718 |
| 7BR | Brusio | 126780 | 807070 | 800 |
| DEH | Degersheim | 247600 | 732600 | 830 |
| SON | Sonogno | 134050 | 703640 | 925 |
| WHA | Wildhaus | 229570 | 746130 | 1000 |
| APT | Alpthal | 212930 | 696860 | 1031 |
| AIR | Airolo | 153400 | 688910 | 1139 |
| 1LC | LaComballaz | 136580 | 572640 | 1360 |
| 4MS | Muenster | 148900 | 663420 | 1410 |
| 7ZN | Zernez | 175259 | 802751 | 1475 |
| 5DF | DavosFluelastr | 187400 | 783800 | 1560 |
| 6SB | SanBernardino | 147290 | 734110 | 1640 |
| YBR2 | Ybrig | 210311 | 705399 | 1701 |
| 7ZU | Zuoz | 164590 | 793350 | 1710 |
| 7SD | Samedan | 156400 | 786210 | 1750 |
| ARO | Arosa | 183320 | 770730 | 1840 |
| LAU2 | LauenenTruettlisbergpass | 141633 | 595482 | 1970 |
| VLS2 | ValsAlpCalasa | 170764 | 735166 | 2064 |
| OBM2 | OberMeielGrossStand | 141183 | 582760 | 2097 |
| FRA2 | FrascoEfra | 132853 | 708906 | 2100 |
| VAL2 | VallasciaSchneestation | 155980 | 690126 | 2268 |
| CMA2 | CrapMasegnSchneestation | 189875 | 733050 | 2330 |
| OFE2 | OfenpassMurtaroel | 168460 | 818233 | 2359 |
| JUL2 | JulierVairana | 149949 | 773049 | 2426 |
| DAV3 | DavosHanengretji | 184616 | 778292 | 2455 |
| TRU2 | TrubelbodenSchneestation | 135519 | 611306 | 2459 |
| 5WJ | Weissfluhjoch | 189230 | 780845 | 2540 |
| DAV2 | DavosBaerentaelli | 174726 | 782062 | 2558 |
| ZNZ2 | ZernezPuelschezza | 175078 | 797312 | 2677 |
| LAG2 | PizLagrevSchneestation | 147050 | 777150 | 2730 |
| GOR2 | GornergratSchneestation | 92900 | 626700 | 2950 |

**Table A1.** Name, location and elevation of all snow station locations used in this study.



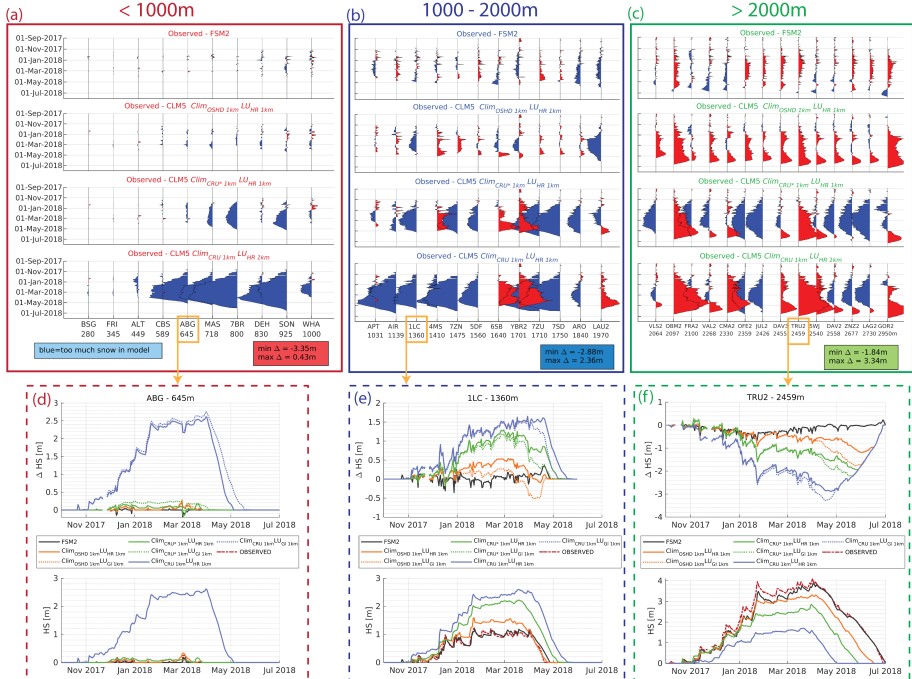

**Figure A1.** a-c feature wiggle plots, which visualize the absolute difference in snow depth between observations and FSM2, CLM5 Clim$_{OSHD1km}$+LU$_{HR1km}$, CLM5 Clim$_{CRU*1km}$+LU$_{HR1km}$ and CLM5 Clim$_{CRU1km}$+LU$_{HR1km}$ simulations, for a selection of stations at elevations lower than 1000m (a), between 1000m and 2000m (b) and above 2000m (c). It is apparent across all elevation bands that FSM2 simulations match observations the closest, and that CLM5 forced with 1km OSHD data and based on a high-resolution land surface dataset is the next best. CLM5 with global meteorological forcing data (Clim$_{CRU1km}$+LU$_{HR1km}$) performs poorly with regards to modelling seasonal snow development, with maximal errors of over 3m, but model performance is improved when using the down-scaled global meteorological dataset to obtain meteorological input data (Clim$_{CRU*1km}$+LU$_{HR1km}$) with particularly dramatic improvements at low elevations. d-f each focus on one station and show the absolute difference to observations as well as seasonal snow depth development of the respective model runs. In addition to the 3 CLM5 configurations shown in a-c, in the first row of d-f, we also show CLM5 Clim$_{OSHD1km}$, Clim$_{CRU*1km}$ and Clim$_{CRU1km}$ with global land surface information (LU$_{Gl1km}$). For these 3 selected examples, the HighRes case performs better for the low and high station location all 3 shown station locations, whereas the global case shows slightly better performance during the melt period for the mid-elevation station. Ultimately, it can be seen though that the effect of meteorological forcing data is substantially larger in comparison to differences arising from the choice of land surface information.



## Appendix B: Point-scale CLM5 validation at FLUXNET stations

The FLUXNET network (https://fluxnet.fluxdata.org/, Pastorello et al. (2020)) provides observations of ecosystem carbon,
water, and energy fluxes at sites across the globe. A total of 6 FLUXNET site locations fall within our model domain and
overlap with our modeling timespan, including a mixed forest, a coniferous forest, alpine and lowland grasslands as well as a
crop site. This analysis is placed in the supplementary material since 4 out of the 6 FLUXNET station locations were lower
than 1000m in elevation, and hence not within the nival zone, preventing an extensive investigation of the link between snow
cover and ecophysiological parameters at these locations. Additionally, the sensor fetch of the 35m high CH-Dav as well as
the CH-Lae station towers integrates a large, heterogeneous area in complex terrain (e.g. including houses etc. for the CH-Dav
site) with highly variable winds depending on the time of the day. Nevertheless we believe this analysis might still be of interest
to some readers, as it validates our PTCLM simulations from a ecophysiological perspective.

Details on the sites, including information on how the prevalent vegetation types translated into CLM5 plant functional
types can be found in Table B1. Observational data for the FLUXNET tower locations were acquired from the European
research infrastructure Integrated Carbon Observation System (ICOS) data product (Team and Centre, 2022), and consists
of standardized observations at half-hourly temporal resolution. Gaps in the data were filled and data was quality controlled
according to the FLUXNET data processing protocol (Pastorello et al., 2020). We focus on the effects of spatial resolution and
meteorological forcing data as we compare CLM5 simulations forced with 0.5°and 1km of $\text{Clim}_{\text{CRU}}$, $\text{Clim}_{\text{CRU*}}$, and $\text{Clim}_{\text{OSHD}}$.

| Site | Name | Lon. (°E) | Lat. (°N) | Elev (m a.s.l) | Site characteristics | CLM5 prescribed vegetation type |
|---|---|---|---|---|---|---|
| CH-Aws | Alp Weissenstein, Canton Graubünden | 9.790639 | 46.583056 | 1978 | Managed (grazed) alpine grassland | 100% C3 arctic grass |
| CH-Cha | Chamau, Canton Zug | 8.41044 | 47.210222 | 393 | Intensively managed grassland | 100% C3 grass |
| CH-Dav | Seehornwald Davos, Canton Graubünden | 9.855972 | 46.815333 | 1639 | Coniferous forest | 100% Needleleaf evergreen tree – boreal |
| CH-Fru | Früebüel, Canton Zug | 8.537778 | 47.115833 | 982 | Medium intensively managed grassland | 100% C3 grass |
| CH-Lae | Lägeren, Canton Aargau | 8.364389 | 47.478333 | 689 | Mixed forest | 50% Needleleaf evergreen tree – boreal |
| | | | | | | 50% Broadleaf decdidious tree – boreal |
| CH-Oe2 | Oensingen, Canton Solothurn | 7.73375 | 47.286417 | 452 | Cropland | 100% C3 Unmanaged Rainfed Crop |

**Table B1.** Name, Location, site characteristics and the selected CLM5 plant functional type for each FLUXNET site used for model performance evaluation.

The FLUXNET tower sites were used to evaluate the performance of the various CLM5 configurations regarding evapotranspiration (latent heat flux) and ecosystem carbon balance (gross primary production). For each FLUXNET site in Switzerland,
we used the absolute error over all time-steps between observations and CLM5 simulations for (a) latent heat flux (LH), and (b)
gross primary production (GPP). Additionally, we perform a one-way analysis of variance (ANOVA) to test for differences in
absolute error between simulation results using different spatial resolution forcings. To test for the significance in differences
in absolute errors we further performed a Tukey's Honestly Significant Difference (HSD) post-hoc test (Abdi and Williams,
2010) for each FLUXNET location. Additionally, in order to investigate significance across all sites we fitted a linear mixed
effects model (Bates et al., 2015) with absolute error of either LH or GPP as a response and the tower site location as random
effects.



Generally, performance differences between the various CLM5 simulations are small (Figure B1), especially when compared to the pronounced effects for the snow-cover development shown in Figure 2 in the main manuscript. However, an ANOVA reported p-values <0.001 for LH and GPP at all sites, revealing significant differences in performance (absolute error) means

between CLM5 configurations. A Tukey post-hoc test confirmed this (Figure B2).

For LH, we see small improvements when using OSHD-based input data at five out of the six locations, while at CH-Cha a marginal decrease in performance when using $Clim_{OSHD}$ compared to $Clim_{CRU}$ is noticeable (Figure B1). However, a linear mixed effects model to assess performance differences between the different CLM5 simulations revealed a significant increase in performance with regards to latent heat flux when moving from $Clim_{CRU}$ over $Clim_{CRU*}$ to $Clim_{OSHD}$ (plot below boxplot

in Figure B1a), whereby performance was further slightly enhanced when using 1km rather than 0.5°forcing data (effect of resolution). Error in GPP simulations showed little variation with the different resolutions and meteorological input datasets, including overlapping extents of the confidence intervals between the different configurations.



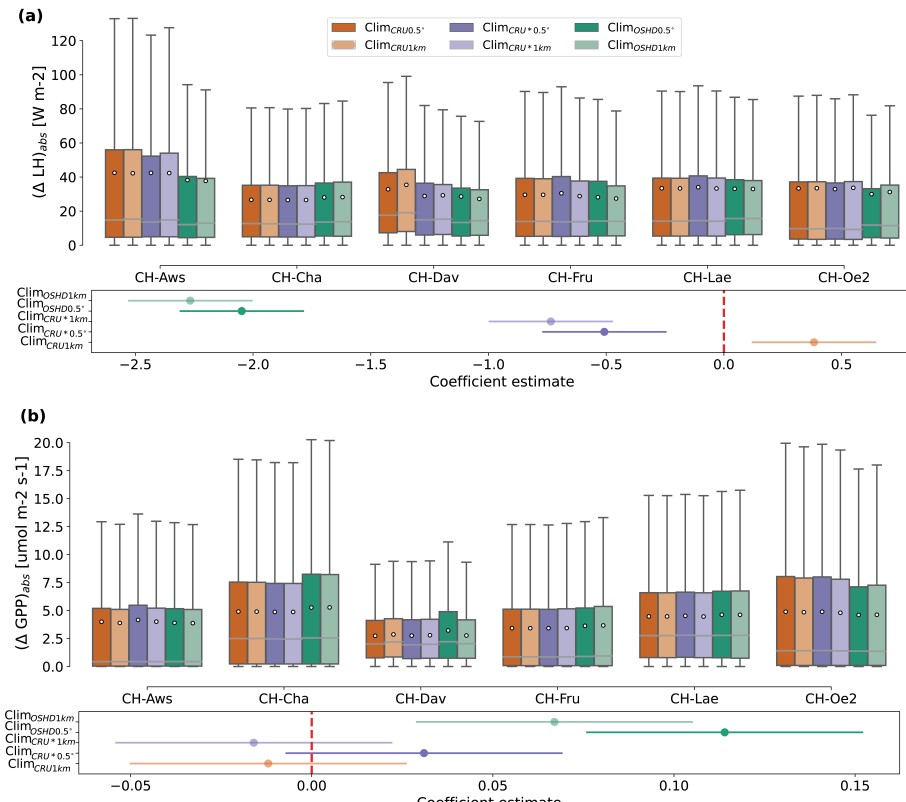

**Figure B1.** Direct comparisons of FLUXNET observations and CLM5 simulations of (a) latent heat flux and (b) gross primary production at six tower locations within Switzerland (see Table B1 for details on the respective sites). Gray lines in the boxplots indicate median error, while mean error is shown with white dots. Plots below the boxplots show the coefficient estimates of a linear mixed effects model with absolute error as response, the various CLM5 configurations ($\mathrm{Clim}_{CRU0.5°}$, $\mathrm{Clim}_{CRU1km}$, $\mathrm{Clim}_{CRU*0.5°}$, $\mathrm{Clim}_{CRU*1km}$, $\mathrm{Clim}_{OSHD0.5°}$ $\mathrm{Clim}_{OSHD1km}$) as predictor, and the site location (CH-Aws, CH-Cha, CH-Dav, CH-Fru, CH-Lae, CH-Oe2) as random effects. Coefficients are in relation to the performance of $\mathrm{Clim}_{CRU0.5°}$, whereby negative values indicate an increase in performance and positive values indicate a decrease in performance. Extent of lines indicates the confidence interval (with a likelihood of 95%).





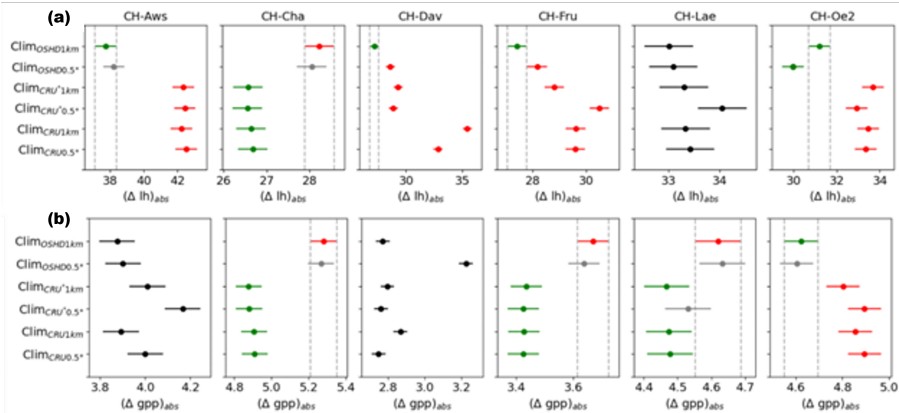

**Figure B2.** The ANOVA reported p-values <0.001 for latent heat as well as for gross primary production, indicating differences between the CLM5 simulations, hence a Tukey's post-hoc test was performed to investigate more into the differences. Tukey's post-hoc test results: Multiple comparisons at all 6 FLUXNET sites between different CLM5 configurations. Here we focus the comparison of the $\text{Clim}_{\text{OSHD1km}}$ configuration ('best-case' after the snow-evaluation) to all remaining ones ('effect of meteorological forcing'). We show absolute differences to observations (FLUXNET towers) across all time-steps for (a) latent heat and (b) gross primary production. Dots indicate mean absolute errors and extents of each line show the confidence intervals (95%); any overlaps indicate a non-significant difference between CLM5 simulations (grey/black dots and lines). Green indicates a significant improvement when using OSHD-based forcing data, red indicates a worsening in performance. Plots with only black indicate a non-significant difference in performance between OSHD-based and CRU-based CLM5 simulations.



## Appendix C: Spatially distributed CLM5 model simulations

This section shows supporting analyses for the spatially distributed CLM5 model simulations presented in the main part of the manuscript.

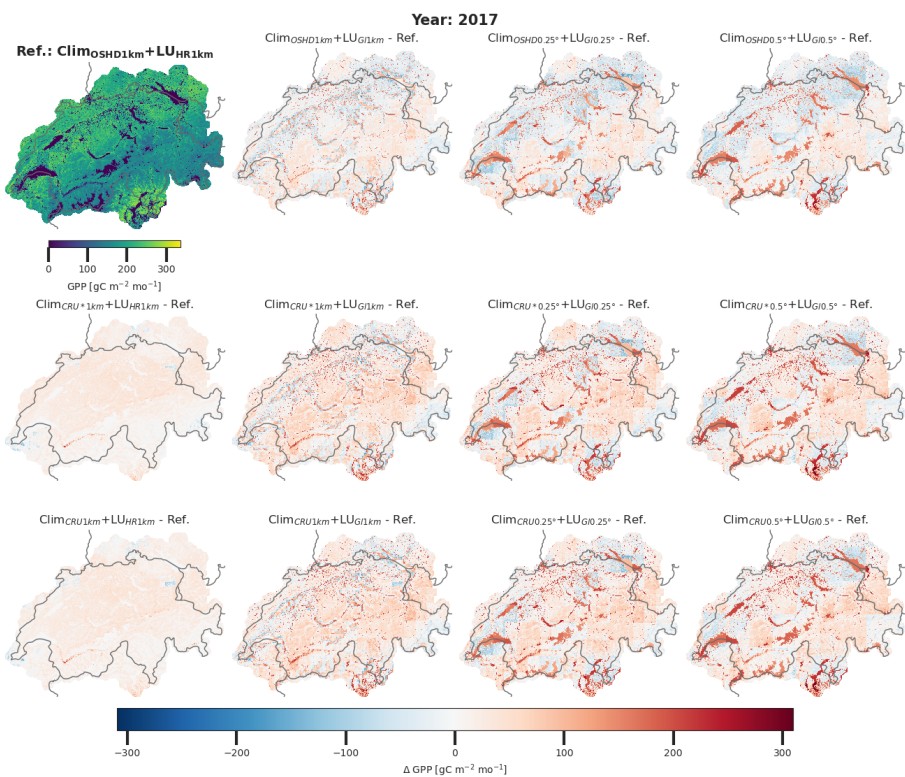

**Figure C1.** Spatial comparison of monthly-averaged gross primary production (GPP) during July and August 2017: The reference case (Clim$_{OSHD1km}$+LU$_{HR1km}$) is compared with simulations of all other CLM5 configurations used in this study. For the residual plots, blue indicates underestimation and red indicates overestimation with regards to the reference case.


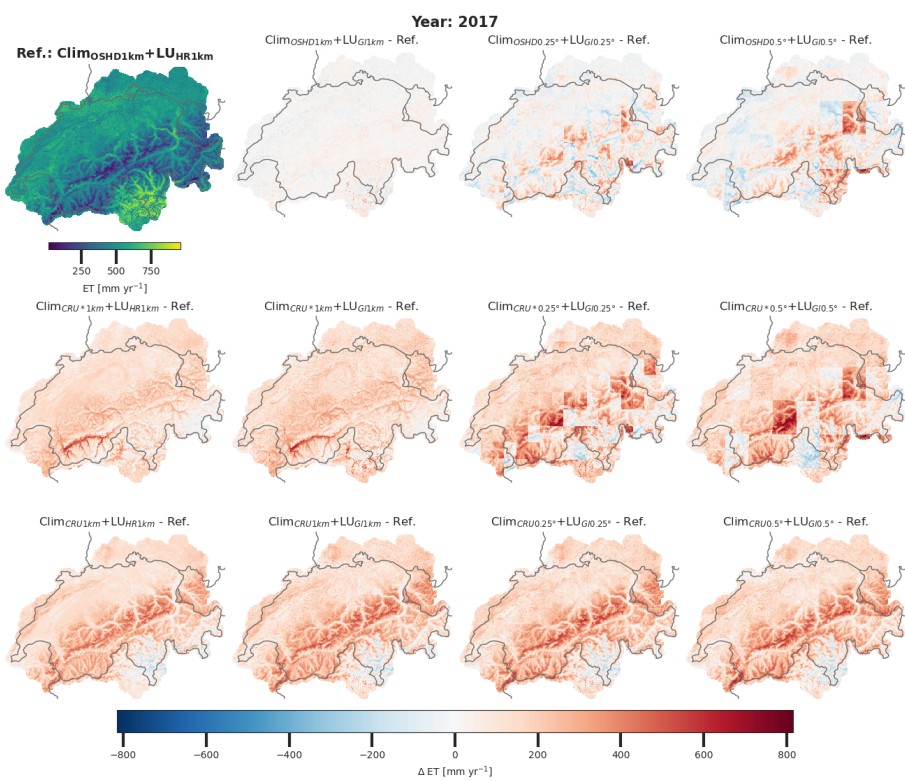

**Figure C2.** Spatial comparison of total Evapotranspiration (ET) during the calendar year 2017: The reference case (Clim$_{OSHD1km}$+LU$_{HR1km}$) is compared with simulations of all other CLM5 configurations used in this study. For the residual plots, blue indicates underestimation and red indicates overestimation with regards to the reference case.

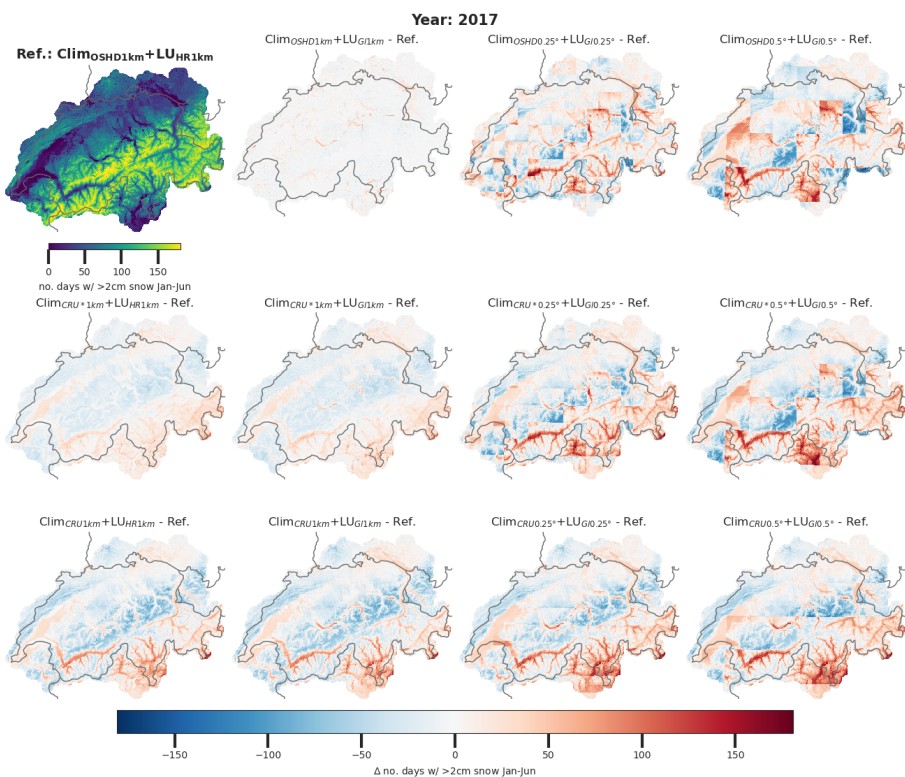

**Figure C3.** Spatial comparison of number of days with more than 2cm snow on the ground between January and June 2017: The reference case (Clim$_{OSHD1km}$+LU$_{HR1km}$) is compared with simulations of all other CLM5 configurations used in this study. For the residual plots, blue indicates underestimation and red indicates overestimation with regards to the reference case.





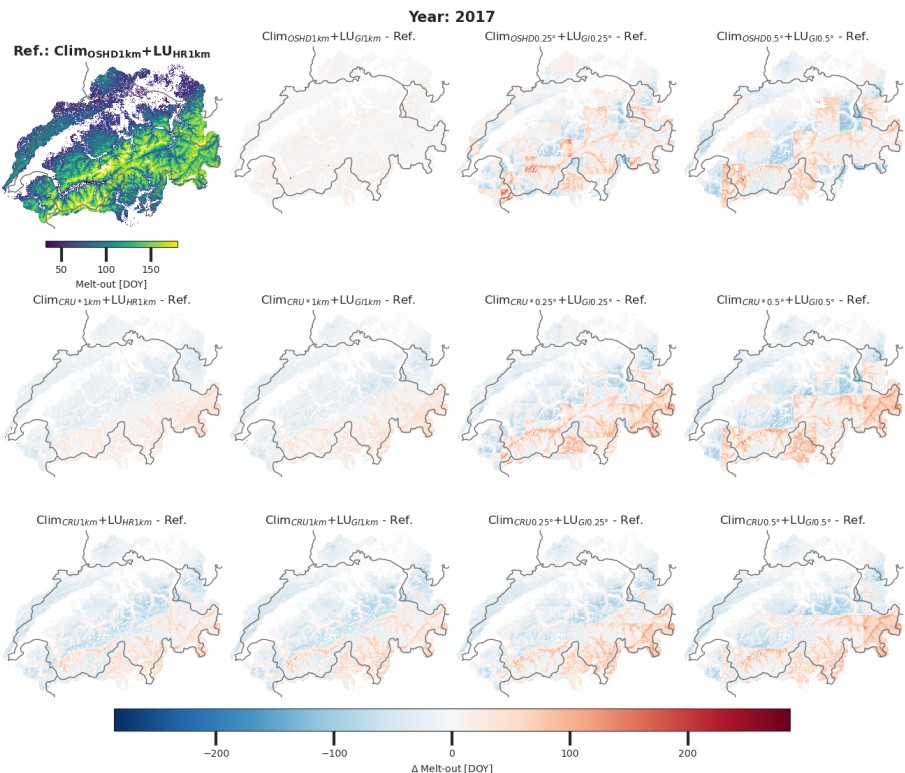

**Figure C4.** Spatial comparison of melt-out date (day of year) during 2017: The reference case (Clim$_{OSHD1km}$+LU$_{HR1km}$) is compared with simulations of all other CLM5 configurations used in this study. For the residual plots, blue indicates underestimation and red indicates overestimation with regards to the reference case.

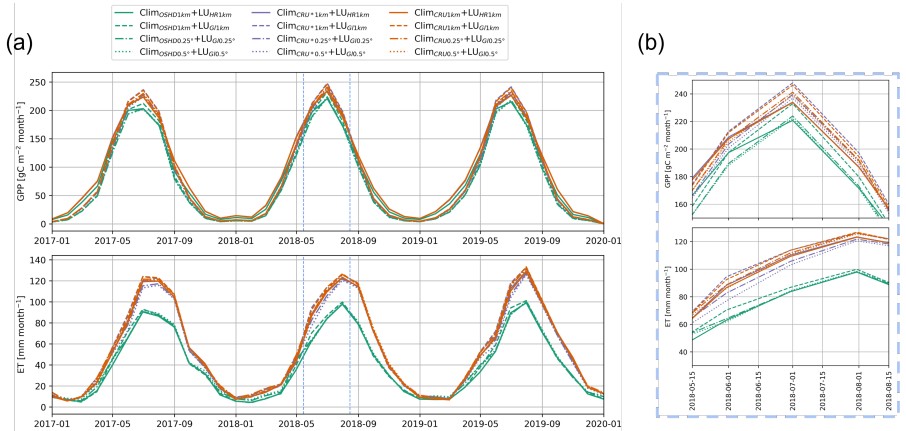

**Figure C5.** Comparison of monthly gross primary production (GPP) and monthly Evapotranspiration (ET) spatially averaged across model domain for all pixels below 2000m and for all 12 CLM5 model configurations. (a) shows 3 yearly cycles between 2017 and 2020, and (b) zooms into the 2018 peak growing season period (dashed blue vertical lines in a).



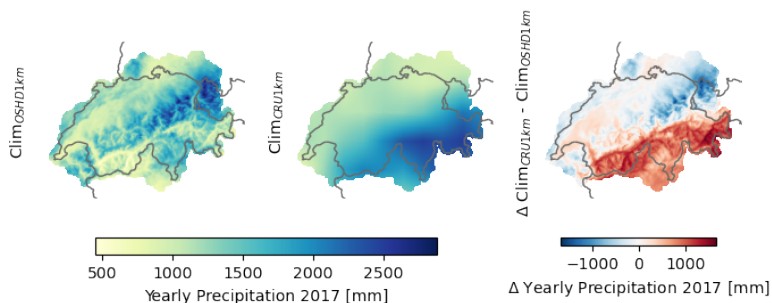

**Figure C6.** Total yearly precipitation input for the year 2017: OSHD-based, CRUJRA-based and a differential plot.

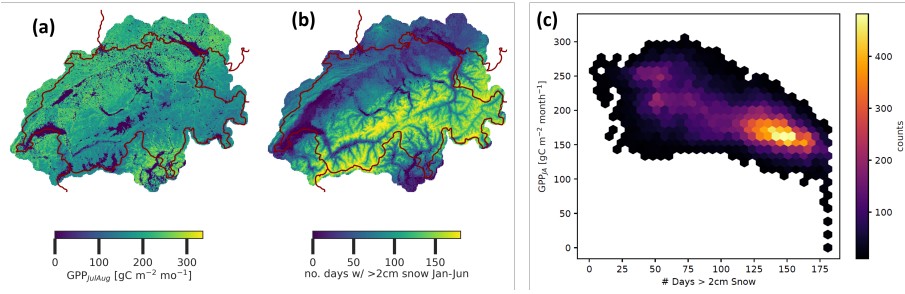

**Figure C7.** Spatial plot of a) monthly-averaged GPP in July and August 2017 and b) number of days with more than 2cm of snow between January and July 2017 as simulated with our best-effort Clim$_{OSHD1km}$+LU$_{HR1km}$ simulation. c) Correlation between number of days with more than 2cm of snow between January and July 2017 and monthly-averaged GPP in July and August 2017 as simulated with our best-effort Clim$_{OSHD1km}$+LU$_{HR1km}$ simulation. Looking at vegetated areas across our entire modelling domain, we see that an increased number in days with more than 2cm of snow on the ground is negatively correlated with peak growing season GPP.



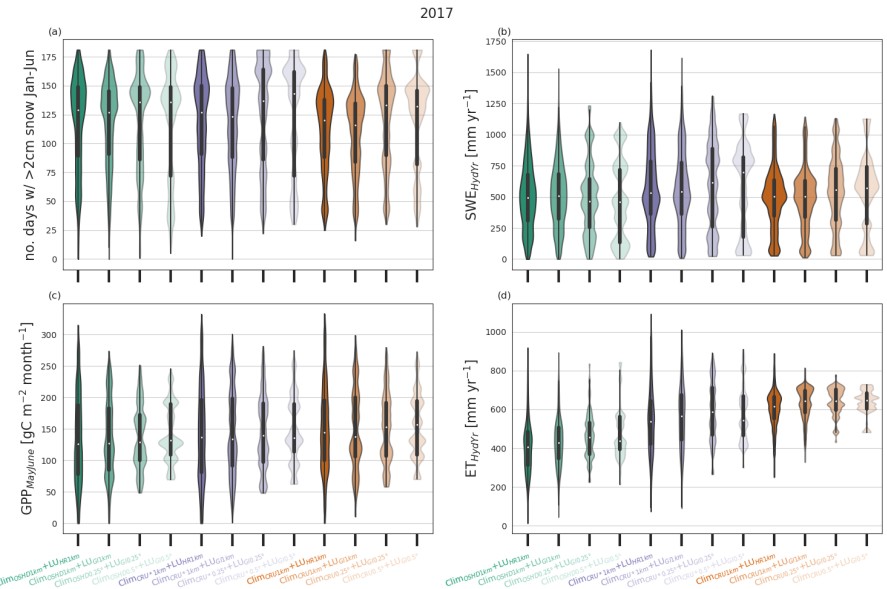

**Figure C8.** Violin-plots showing comparison of all 12 CLM5 model configurations for the year 2017 across the entire model domain: (a) number of days with >2cm of snow between January and June 2017, (b) cumulative SWE (total positive SWE increments; 'how much water is stored in total') during the hydrological year 2017 (1.10.2016 - 30.09.2017), (c) monthly-averaged GPP during May and June 2017and (d) total Evapotranspiration during the 2017 hydrological year. In addition to information obtained from a box plot ($25^{th}$ + $75^{th}$ percentiles and median), the violin plots show a kernel density estimate of the data.

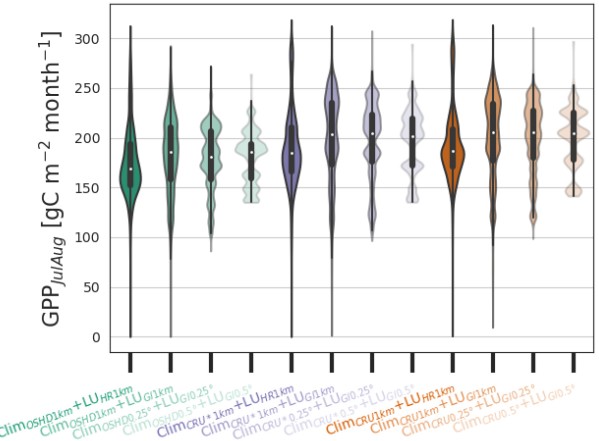

**Figure C9.** Violin plot showing distribution of all 12 CLM5 configurations across the entire model domain: monthly-averaged GPP during July and August 2017.



*Author contributions.* All authors helped design the experiments. JM set up the modelling infrastructure and performed the CLM5 simulations. JM performed the analysis, with input from all authors. JM wrote the manuscript, with contributions and feedback from all authors.

*Competing interests.* The authors declare that they have no conflict of interest.

*Acknowledgements.* The authors thank Thomas Kramer and his team for HPC support throughout this project. JM received funding from
a WSL internal project call. DNK and JM were supported by the Swiss National Science Foundation SNF (project: Adohris, 205530), as was GM (project: P500PN, 202741). We further thank the team of the operational snow hydrologic service at SLF for providing input data. Developers of open source python toolboxes, particularly xarray (Hoyer and Hamman, 2017) and xesmf (Zhuang et al., 2023), have also played a crucial role in this study by enabling efficient analysis and manipulation of large datasets.



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
