# Peer review of "Regionally optimized high resolution input datasets enhance the representation of snow cover and ecophysiological processes in CLM5"

_EGUsphere, 2023_

## Author Comment (AC1)

We thank the reviewer for taking the time to read through our paper so carefully, for their detailed review and for their insightful comments. Please find our replies below as inserted in red text.

**General comments**

The paper by T. Malle et al. addresses the effects of spatial resolution, atmospheric forcing data, and land surface characterisation on snow depth, gross primary production (GPP) and evapotranspiration (ET) simulated by the Community Land Model version 5 (CLM5) over Switzerland. Factorial model experiments with different combinations of resolution and input data were used to isolate the effects of the three sources of variability in the model output. The authors conclude that deficiencies in all three aspects contribute to the current uncertainty in the results of land surface models (LSMs), particularly in heterogeneous regions like the Swiss alps. In this context, they call for the use of more fine-grained input data and for evaluating LSM simulations at high resolution.

The study contains several elements that are interesting for the audience of ESD, including modelers and users of land/climate model output over heterogeneous regions. It probably involved huge efforts, considering the generation of various input datasets, model experiments, the analysis of three simulated quantities (snow depth, GPP, ET), and their comparison against observational or modelled benchmarks. For a high-quality modelling paper, I think the technical foundations need to be improved (e.g. what controls simulated snow depth, GPP, and ET in CLM5-SP offline?). Also, the study addresses several complex issues in land/climate modelling (e.g. ecophysiology, carbon cycle dynamics, feedbacks, etc.) that are poorly captured in the model setup chosen for this study (i.e. CLM5 with prescribed vegetation phenology, largely inactive carbon and nitrogen cycles, and prescribed atmospheric conditions). I think this distracts from the relevant key findings and confuses the overall storyline. Below, I list suggestions how the study and its findings could be improved to become more useful for the audience of ESD. I think this involves substantial revisions. My comments refer to main text only, not the Appendix.

Thank you for this comment, and for the generally positive assessment of our work. Our goal is to maximize the paper's usefulness for the modelling community, so we very much welcome your suggestions for improvement.

Initially, we opted to use CLM5 with prescribed vegetation phenology to represent the vegetation of Switzerland today in the most accurate manner, which would be very difficult to achieve when running in prognostic biogeochemical mode, at least without data assimilation. We also did/do not have the computational resources to perform CLM5 simulations at high resolution in bgc mode, especially not the required spin-up. We will therefore follow your suggestion of re-focusing the manuscript on snow-related analysis and will not focus on the link between snow and GPP/ET.

We will elaborate in our response to the specific comments below.

**Specific comments**
**(1) Relevance of higher resolution and implications of the results**. It is obvious that higher resolution and more specific input data improve model performance for a specific location, i.e. when CLM5 output is evaluated against high-resolution simulations or point/site-scale observations. This is known by the modelling community and it is thus not surprising that the 1km simulations with high native-resolution input data perform best. What additional insights can the study provide, e.g. should we focus on (1) high resolution atmospheric forcing data in offline land surface modelling, (2) high resolution land surface data, or (3) high LSM resolution to capture non-linear effects between atmospheric forcing and land surface characterization? Or should model evaluation and benchmarking be done at high resolution, to rule out deficiencies due to insufficient resolution and focus on process uncertainties? What do users get if they run CLM5 at high (<0.5°) resolution with CRU as the atmospheric forcing dataset, is CRU 0.5° simply interpolated and could "smart downscaling" considering the temperature lapse rate be useful as a (built-in) solution? (this would be very interesting for the CLM community!)
We will discuss our results in this context in greater detail in a revised version of the manuscript.

**(2) Benchmarking and implications**. The purpose of a land surface model like CLM5 is to simulate larger spatial scales ranging from one grid cell to regional domains or the globe, in spatially representative grid cells rather than point fashion (only at very high resolution, grid cells start to resemble point/site-scale conditions…). Therefore, it might be useful to evaluate the output of various simulations (1km, 0.25°, 0.5°) at the lowest resolution of 0.5°, to assess if there are major differences due to input data quality and non-linearities. Such results could inform the modelling community if there is a need to account for small-scale heterogeneity to obtain accurate fluxes and pools at larger spatial scales of one to several kilometres. This might ultimately be more relevant for the purpose of CLM5 (or LSMs in general) than mimicking point/site-scale conditions.
This is a very valid point. We have redone evaluations of gridded snow simulations at 0.25°, as we believe that given the complexity of the topography across our modelling domain and its relatively small size, and considering today's ever-increasing computational resources, 0.25° should be a fair target for the main analysis.

We have upscaled results from the 1km simulations to 0.25°, using a conservative upscaling approach which preserves areal averages. For this purpose, we had to decrease our evaluation domain slightly, as we performed the 1km simulations with a mask running exactly along the edges of our modelling domain, making it impossible to upscale these areas to 0.25° without crude assumptions. The 0.5° simulations were downscaled to 0.25°, and all simulations were evaluated across the same domain.

As seen from the updated version of the Taylor's diagram below (Figure 3 in original manuscript), the difference between different land-use datasets with regards to simulated snow depth remains small and increasing spatial resolution in isolation only has a marginal effect on accuracy of simulated seasonal snow cover. Upscaled 1km simulations with highest quality meteorological forcing datasets (Clim$_{OSHD1km}$) perform best during the early accumulation period but performance is matched/exceeded by the lapse-rate corrected global dataset (Clim$_{CRU*1km}$) for mid-accumulation and ablation period, underlining the effect of a relatively simple lapse-rate

based downscaled temperature input to better account for sub grid variability. We will discuss these results in greater detail in an updated version of the manuscript.

[Figure]

**(3) Land surface dataset methodology and evaluation**. The generation of a high-resolution land surface dataset based on national land cover data and satellite-based LAI (not sure I interpreted the brief description in section 2.3.2 correctly) seems quite innovative and interesting. I think the procedures should be described in more detail, so that other could potentially follow a similar approach. Some summary/evaluation/validation beyond what is shown in Figure 1 (e.g. how did land cover fractions and LAI change across Switzerland, regionally, or for the point locations?).
We will describe how the land surface dataset was created in a more detailed manner. Additionally, we will add additional figures to the appendix demonstrating how different percentile fractions (e.g. crop, vegetation) and LAI/SAI changed spatiotemporally across the model domain.

**(4) Overall scope including snow depth and GPP/ET**. The manuscript might benefit from limiting the scope and analysis to snow depth, and in that area developing the causes for improved model performance more thoroughly (e.g. how can spatial resolution, atmospheric forcing data, and land surface characterization influence snow depth considering forced precipitation, rain/snow partitioning inside CLM5, land cover, LAI, slope, etc.). A caveat of this is, of course, that snow depth is likely closely linked to the forced precipitation and temperature fields, which could make the results appear trivial. Yet, the authors might identify interesting aspects related to, e.g., land cover, LAI or slope. In any case, there are multiple reasons why the analysis of effects on GPP and ET should be excluded, or included only after substantial revisions:

a) The link between Hypothesis 2 and the chosen methods is currently very weak. Most importantly, the modelling setup and the correlation analysis do not allow to isolate the effects of differences in snow depth from differences in its drivers (i.e. spatial resolution, atmospheric forcing data, and land surface characterization) on GPP and ET. The framing of GPP and ET as "snow-cover dependent ecophysiological variables" is confusing, because likely most of the GPP and ET differences are driven by the resolution-dependent forcing fields directly (i.e. atmospheric variables or land surface characterization) and not indirectly via snow-cover changes. This joint independent driver could lead exactly to the correlation between snow cover and GPP/ET found by the authors, without any effect of snow cover itself on GPP/ET.

b) In CLM5 in SP mode, vegetation phenology is prescribed as a climatological seasonal cycle of LAI, and LAI controls the leaf to canopy scaling of all fluxes including carbon (GPP) and water (ET). Therefore, in SP mode the model has very limited capabilities to show an ecophysiological response to snow cover change, in the sense of seasonally shifting growth. As a minimum, I recommend to discuss the precise implications (e.g. is the response we see solely due to changes in temperature and water availability, or what can affect GPP in SP mode at all?). However, I strongly doubt that an ecophysiological response can be quantified in SP mode. I imagine it works as follows, although I am not 100% sure: if the snow season is shorter than in the climatology, suitable temperatures for growth coincide with zero LAI in the model and nothing happens until the climatological growth begins; if the snow season is longer than in the climatology, suitable temperatures for growth coincide with high LAI in the model, leading to a jump start and no compensation later in the season.

c) Ecophysiology refers specifically to (plant) organisms and not to land as a whole. So, for ecophysiological effects on ET I suggest analyzing plant transpiration and canopy evaporation fields of CLM5, or total ET in the vegetated parts of grid cells excluding the bare soil PFT. However, considering that the experiments include changes in land surface characterisation, the best option might be to remove the term "ecophysiological" and to refer to responses in land ET, which include e.g. the effects of varying bare soil fraction.

d) Simulated GPP and ET are compared against the "best-effort" configuration of CLM5 at 1km resolution as a benchmark, and deviations from this benchmark are considered model uncertainty. An objective benchmark (e.g. observational data or output of a dedicated model, like for snow depth) is lacking. Also, I am not sure the term "model uncertainty" is appropriate in this context. Is it model uncertainty if lower resolution models perform poorer at producing high-resolution-model-like outputs? The comparison could potentially be done the other direction (see point 2).

Thanks for the suggestions and elaborate explanations. We will follow the suggestion of re-focusing the manuscript on snow depth and will not focus on the link between snow and GPP/ET. We will discuss rain/snow partitioning in more detail and include an analysis of snow-depth accuracy per elevational band. Figure A1 will be moved to the main paper and Appendix B will be removed entirely.

We would like to keep Figure 4 in the paper, however, not as a main result, but in order to demonstrate how different GPP/ET results can be in our various simulation setups. It will be integrated as part of the discussion and help pointing at the magnitude of uncertainties, which are related to questions like resolution, input data, and resolving vs. not resolving sub grid variability.

**(5) Introduction**: The background could be more technical (e.g. what controls simulated snow depth, GPP, and ET in CLM5-SP offline?) and focused on aspects that matter for this study (i.e. strengthen the main story and cut out things that are interesting but not directly relevant).
We will include a more technical paragraph in the introduction of a revised manuscript.

**(6) Discussion**: The findings need to be contextualised, considering the capabilities of CLM5 in SP mode offline (see above). For the comparison with other studies (e.g. Birch et al.) to be valid and useful for the reader, it would be good to understand what these studies did or how they explain their biases (e.g. also land cover specification, atmospheric forcing and model resolution, or something completely different like effects of grazing animals or artic plant types?). Certain

links made between the findings of this study (with a very specific setup and research focus) and model uncertainties (with sometimes known different root causes) are not appropriate (e.g. L371). I recommend reconsidering these and focusing the discussion more on new insights gained through this study (see points 1 and 2).

We will revise the discussion and focus it more on the highlights of this study.

**(7) Language and figures**: The manuscript is very well written. There are a couple of "empty" phrases that highlight something but do not actually deliver new insights (e.g. the results have profound implications, the study highlights the importance of model development, the study highlights the utility of multi-resolution modelling, etc.). I think those could be filled with content or removed. The figures have high quality and are visually appealing.

Thank you. We will remove empty phrases from the paper.

**Technical and line-by-line comments**

L14 ff: "Earth's systems" sounds a bit unconventional and far-reaching; maybe use a more concrete/narrow term?

We will rephrase this sentence.

L19: add water ok

L23, L24: maybe use "influence" instead of "control/determine" ok

L24, L35, L65: there is no feedback among the mentioned dependencies/effects

We will use 'exchange' instead.

L29, L31: check logical link for "thus" and "as" ok

L49: offer literature for multi-resolution modelling?

We will cite Singh et al. (2021, https://doi.org/10.1002/2014WR015686) and Meissner et al. (2009, https://doi.org/10.1127/0941-2948/2009/0400 ) here.

L50: they allow evaluating ok

L51: I guess any gridded LSM can be evaluated against point/site-scale observations by taking the individual grid cells that match the locations best; there is no need for dedicated point simulations for this

In principle yes, but for this study we used downscaled meteorological forcing data to the exact location of the point observation for our '1km' simulations, which in case of the OSHD dataset also included station observations and should represent a best-case scenario. Also, for the lapse-rate based temperature correction we used the exact coordinate and a high-quality GPS elevation measurement for the downscaling. We will make this clearer in the methods section of a revised version of the manuscript. We also acknowledge that our choice of naming of the point simulations might have fueled this confusion, which we will therefore update in a revised version of this work (e.g. simply $Clim_{OSHD}$ and $Clim_{CRU*}$ instead of $Clim_{OSHD1km}$ and $Clim_{CRU*1km}$) to make this clearer.

L56: what is meant by snow cover "dynamics"? the temporal evolution?
Here we mean depth, duration as well as variability of snow cover across space and time. We will make this sentence clearer.

L73: consider limitations in SP mode ok

L80: in my understanding, "process representation" refers literally to how processes are represented in the model, i.e. the equations used to calculate snow depth, GPP and ET; I think this term is not appropriate for the modifications in forcing/input data and resolution made in this study
We will avoid using the expression 'process representations' in a revised version of the manuscript.

L87: remove "heat fluxes" if not addressed in the results
Heat fluxes will be removed here.

L88-90: consider reformulating, the sentence sounds nice but it does not deliver any content (or at least I cannot understand it)
We will reformulate the sentence.

L98-102: the methods need some technical precision here: which state variables, which datasets (e.g. what time period/ past conditions do they represent), not only "natural" vegetation (crop PFTs), how does GPP work in SP mode (if the LUNA model active, also cite Ali et al. 2016 for photosynthesis)
We will cite Ali et al. 2016 here (LUNA model is active in our simulations) and give more technical background on our simulation setup.

L100: "is approximated": I think this a used choice, not done by the model.
We will rewrite this sentence.

L103: revise components of ET, e.g. soil sublimation does not make sense; maybe "ice" is missing?
Yes, soil should read ice here, we will update it accordingly.

L105-111: snow cover is the focus of this study, so I think the foundations of snow cover calculations should be provided for convenience (and understanding), including rain/snow partitioning in CLM5
We will document and explain CLM5 snow cover calculations including rain/snow partitioning in greater detail here.

L115, L197: PFTs are patch level, mention that prescribed LAI etc. is in SP mode.
We will update this sentence accordingly.

L116-L119: is there any difference between taking out individual grid cells from the regional simulations and running dedicated point simulations (i.e. a 1x1 regional grid) in your setup? they should be identical, provided that (1) the grid is anchored and specified identically, and (2) there are no lateral exchanges between grid cells (which depends on the CLM5 compset, if river routing is off there is no lateral exchange)

There is no lateral exchange in our model setup, so in principle there is no difference in the simulation setup. However, as mentioned above we used downscaled meteorological forcing data to the exact location of the point observation, resulting in different forcing datasets than for the 1km simulations. We will make this clearer in the text.

L124: for comparing between resolutions, it would have made sense to subdivide the 0.5° and 0.25° grids, e.g. by using 4x the number of cells for 2x resolution, i.e. 20x12 grid cells for 0.25°; that way you could preserve the grid anchoring and preclude differences due to a new "positioning" of the grid; maybe motivate your choice and/or mention potential effects of different grids on the results
The 1km grid was pre-determined by the OSHD grid of snow-simulations and meteorological forcing data, which we used as a starting point. The 0.5° and 0.25° grid were then determined to closely match the extent of this initial grid. We will discuss potential effects of this choice on the results in a revised version of the manuscript.

L128: a "the" is missing ok

L137 ff: are the grid cells for point simulations centered on the station coordinates? using the "nearest neighbor" grid cell for land surface characterization seems like a very simplified approach compared to all the other sophisticated things done in this study; why not use conservative regridding so you would get something more realistic? ideally, one would generate surface data from the raw PFT fraction data at 0.05° resolution (CLM's own methods) or the raw Swiss national data (your methods); in contrast, taking the nearest neighbor effectively shifts surface information and pairs it with the correctly positioned atmosphere; if this is acceptable, one might as well take the nearest (or interpolated/regridded) results of the gridded simulation? (see comment on L116-L119)
Yes, the grid cells for point simulations are centered on the station coordinates. We did actually first calculate the domain and surface dataset for each point location separately via CLM's own methods, hence using the raw PFT fraction data at 0.05° resolution. We then updated this dataset with the 1km HighRes dataset taking the nearest neighbor. Given that some of the underlying datasets are higher resolution than 1km we acknowledge that it would be beneficial to also run our own methods at the effective point location. We will prepare this dataset, and since all station locations are open, non-forested sites we will ensure that we have a 100% vegetated non-forested grid cells for all LU$_{HR}$ simulations. We will include these new simulations in a revised version of the manuscript and describe it accordingly in the text. We also acknowledge that we should re-name the point simulations (e.g. simply LU$_{HR}$ and LU$_{Gl}$ instead of LU$_{HR1km}$ and LU$_{G1kml}$) to avoid confusion and make this point clearer.

L154: is "accelerated decomposition" valid/applicable for SP mode? it sounds like BGC; by "cycling" (remove "re-")
You are right, "accelerated decomposition" is not necessary for SP mode, this part of the sentence will be removed. "Re-cycling" will be replaced by "cycling".

Figure 1 caption: is "percentage vegetation cover" the natural vegetation landunit including bare soil, or the sum of vegetation PFTs and CFTs? (the latter would be good)
Currently it is the natural vegetation landunit including bare soil, but we will update this accordingly to represent the sum of vegetation PFTs and CFTs.

L159, L177: is CRU a station-based interpolated dataset and the OSHD based on the COSMO model? for OSHD, it is also a bit unclear if the dataset was produced or re-used for this study
Yes, we will make this clearer in the text of a revised manuscript.

L187: was the native 0.05° PFT and LAI data reprojected and regridded, or was this done based on an existing surface dataset at e.g. 0.5° resolution? depending on the data used, there might be several regridding steps involved (with every step further degenerating the final product) and the native input might be 0.05 or 0.25°; was this done with the CLM5 tools, with which regridding algorithm (bilinear, conservative)?
CLM5 tools were used separately for the 1km, 0.25° and 0.5° resolution to generate each of the 'global' datasets of this study ($LU_{Gl0.5}$, $LU_{Gl0.25}$, $LU_{Gl1km}$), using the conservative regridding algorithm and all underlying raw input data files (e.g. 0.05° raw PFT fraction data).

L200: FSM2 output is not "observational"
We will clarify this.
L216-217: remove "were", "ground truth" is usually used in remote sensing?, does "upscaled" mean regridded and if so, with which algorithm (bilinear, conservative)?
Ok - "were" will be removed, and "ground truth" will be replaced by "best-case reference". Here upscaled means regridded with the conservative algorithm. We will add this information to the sentence.

L230-234: consider doing this at 0.5° resolution, see point 2
See our answer to point 2 earlier.

L238: to be honest I am a bit lost by now – was 3.1.1 done with the point simulation results? maybe for each Results sub-section this could be highlighted in the title or mentioned in the first sentence.
Thanks for this suggestion, we will make the sub-section titles more descriptive.

L247: for this section it would be really good to understand how land cover and LAI and potentially affect snow depth in CLM5 (see point 4)
From our analysis of snow-depth simulations we saw that the choice of surface dataset only had marginal effects on simulated snow cover, which is why we have not included an analysis of the effects of land cover and LAI on snow cover here.

L283: replace "parameters" by "variables" ok

L284: why is peak GPP assessed and not total GPP? I see the "motivation" later in L328, but because there is a bigger effect does not mean it is more relevant? I think this is related to limitations in SP mode (see point 4b); the effects described in section 3.2 go way beyond ecophysiology (see point 4c)
We will update Figure 4 to use show total annual GPP instead of peak GPP. This part of the results section will be rewritten.

Figure 4 labels: replace "climatological" by "meteorological" ok

L322: snow cover and ET are negatively correlated, but I doubt this is driven by snow but rather by cold temperatures and energy (not water) limitation (see point 4a); feedbacks to the atmosphere are missing in CLM5 offline by construction

Following your suggestion in the specific comments we will refrain from discussing these links in the paper. The section 'Seasonal snow cover development and ecophysiological variables' will be omitted from the paper. Instead, we will prepare a more detailed assessment of snow-related simulation results.

L348 ff: for the calculation of variations in (monthly) total GPP across Switzerland, it would be useful to have an observational benchmark and to relate the amounts to total GPP (i.e. % variation of total GPP). Is there a good reason for not calculating variation in total annual GPP? I would find this quantity more informative

We will update Figure 4 to show variations in the total annual GPP. As we shifted this paper's focus, we will not include observational benchmarks but discuss relative differences.

L368: ET can also be water limited in Switzerland, at least in some regions seasonally

We will add this to a revised version of the manuscript.

---

## Author Comment (AC2)

We thank the reviewer for taking the time to read through our paper, for their detailed review and for their insightful comments. Please find our replies below as inserted in red text.

Although land surface modeling has evolved from simple biophysical parameterizations to complex frameworks in recent years, large uncertainties remain especially in mountainous regions and areas with complex terrain. This study uses a multi-resolution modeling setup to investigate the impact of meteorological forcing data, spatial resolution, and land surface data on the simulation of snow cover and ecophysiological variables. The authors perform simulations using the Community Land Model version 5 (CLM5) over Switzerland. They found that increased resolution not only improved the representation of snow cover in CLM5 but also propagated through the model and affecting the gross primary productivity (GPP) and evapotranspiration (ET).

Overall the manuscript is well written and of interest to the land surface/earth system modeling community. However, the CLM5 model setup and the model evaluation method do not appear to be appropriate in the current manuscript. In specific, (i) the CLM5 is setup in prescribed satellite phenology mode (~ fixed growing season), yet one of the main focus of the study is to investigate the link between snow cover duration and growing season length (and GPP/ET); (ii) CLM5 simulations in the study were conducted at three resolutions: 1km, 0.25deg, and 0.5deg, but the evaluations were performed at 1km resolution (Figure 3), which are not fair comparisons in my opinion.

I suggest the authors choose the prognostic biogeochemistry mode for CLM5 simulations, and perform model evaluation at the resolutions of the respective simulations. In addition, I have some minor suggestions outlined in the comments below that will hopefully improve the future version of the paper.

Thank you for this comment.

**Regarding (i):** Initially, we opted to use CLM5 with prescribed vegetation phenology to represent the vegetation of Switzerland today in the most accurate manner, which would be very difficult to achieve when running in prognostic biogeochemical mode, at least without data assimilation. We also did/do not have the computational resources to perform CLM5 simulations at high resolution in bgc mode, especially not the required spin-up. We acknowledge the problems arising from this setup when focusing on the link between snow cover and growing season length and have decided to reframe the paper following a valid suggestion from Reviewer 1. The revised manuscript will be more focused on input data, resolution, and snow, while removing in-depth discussions of links of all that on GPP/ET estimates. For this purpose, we will move Figure A1 from the appendix to the main paper, and further include more detailed performance assessments of the various CLM5 model setups along elevational bands.

**Regarding (ii):** This is a very valid point. We have redone evaluations of gridded snow simulations at 0.25°, as we believe that given the complexity of the topography across our modelling domain and its relatively small size, and considering today's ever-increasing computational resources, 0.25° should be a fair target for the main analysis.

As seen from the updated version of the Taylor's diagram below (Figure 3 in original manuscript), the difference between different land-use datasets with regards to simulated snow depth remains small and increasing spatial resolution in isolation only has a marginal effect on accuracy of simulated seasonal snow cover. Upscaled 1km simulations with highest quality meteorological forcing datasets (Clim$_{OSHD1km}$) perform best during the early accumulation period

but performance is matched/exceeded by the lapse-rate corrected global dataset (Clim$_{CRU*1km}$) for mid-accumulation and ablation period, underlining the effect of a relatively simple lapse-rate based downscaled temperature input to better account for sub grid variability. We will discuss these results in greater detail in an updated version of the manuscript.

[Figure]

Specific comments

L164, the sentence does not read well. We will reformulate this sentence in a revised version of the manuscript.

L169-170, do you assume values in the original dataset (ClimCRU1km) are at sea level, and apply the temperature lapse rate based on the mean elevation of each 1km grid from the CRUJRA data? Which elevation data do you use? I would suggest include a map of elevations in Figure 1. We do not assume that the values in the original dataset are at sea level. Rather, we use a global DEM at 0.5deg to first bring temperature to sea-level temperatures by applying negative lapse rates. We then use a high-resolution DEM of Switzerland at 1km to re-lapse temperature. We will add this information to the methods section of a revised manuscript. We will also include a map of both the high and low resolution DEM to the appendix of a revised version of this manuscript.

L172, it would be helpful to add a description of the snow/rain partitioning method in CLM5. CLM5 partitions total precipitation into rain and snow according to a linear temperature ramp, resulting in all snow below 0 °C, all rain above 2 °C, and a mix of rain and snow for intermediate temperatures. We will include this information to the methods section of a revised manuscript.

L181-182, do you just aggregate the 1km data to 0.25deg and 0.5deg? Please try to describe what exactly is being done. The ClimOSHD forcing data would be useful for other modelers, is the data available?
Yes, here the 1km data was upscaled to 0.25 and 0.5deg using a conservative regridding approach. Thanks for your note, we will consider publishing the Clim$_{OSHD}$ forcing data, but certainly as a separate data paper given the enormous effort required to prepare such datasets Simulation results and updated surface data from this study will be made available as part of this paper.

L234, given that the met forcing and landcover data etc. are all at coarser resolutions, it is not fair to evaluate coarser resolution (0.25deg, 0.5deg) CLM5 simulations using finer resolution (1km)

observations. I suggest the authors regridding the 1km observation data to the 0.25deg and 0.5deg first, then redo the comparisons and Figure 3.

As mentioned in our comment above, we have followed your advice and redone the evaluation at 0.25deg, which we believe is a fair target.

L241-245, it would be helpful to show or discuss which variables in the met forcing data contribute to the different CLM5 simulations.

We will include a section in a revised version of the manuscript where we discuss and show differences in meteorological forcing variables between the various datasets.

Figure 3 is an important figure in the paper, but the Taylor plots and labels/legend are too small, and hard to read.

Thanks for pointing this out, we will improve the quality of the Taylor plots.

In the captions of all the figures, a summary of main results is also included, which is not necessary and makes the captions too long.

We will refrain from giving a summary of our results in the figure captions.

L256, supplementary material is not found.

This was a mistake and should read Figure A1, thanks for spotting it. Figure A1 will be moved to the main manuscript in a revised version of the manuscript.

L265, Figure 3 needs to be cited here.

Yes, we will cite Figure 3 here.

L273-275, I suggest the authors redo these evaluations at the resolutions used for each CLM5 simulations.

We have redone all evaluations at 0.25deg, which we believe is a fair target for our model analysis.

L290-291, the sentence doesn't read well.

We will reformulate this sentence.

Figure 4, note the 3$^{rd}$ panel are labeled as effect of climatological forcing instead of meteorological forcing.

Thanks for catching this, we will update the label.

L316-317, the sentence does not read well.

We will omit this section from a revised version of the manuscript.

---

## Author Comment (AC3)

We thank the reviewer for taking the time to read through our paper and for their insightful comments. Please find our replies below as inserted in red text.

The manuscript by Malle et al. investigates the impact of spatial resolution, quality of atmospheric forcing datasets and land-use information on the simulated snow depth, GPP and ET over the spatial extent of Switzerland and adjacent watersheds of neighboring countries by using the Community Land Model 5 (CLM5). Simulations of different combinations of meteorological forcing and land-use information were conducted to analyze changes in model performance. In addition, CLM5 simulated snow depth were compared with station observations and results from a spatially distributed, physics-based snow model. The authors find the combination of increased spatial resolution of model and high-quality input datasets can improve the representation of snow cover in CLM5, and these improvements further propagate through the model, directly affecting GPP and ET. The manuscript demonstrates the importance of high spatial resolution and high quality input datasets for climate impact studies.

The manuscript dedicated a detailed description of methodology, but the explanation of the results is somewhat brief, and most of them are descriptive, lacking of model processes related analysis and discussion. Such as, what controls the snow depth simulation in CLM5, how the different forcing datasets affect snow simulation? how the improvements in snow propagate in CLM5 in a cascade way, what's the linkage between snow cover and GPP and ET. I suggest the authors improve these parts. In addition, the figures in the manuscript should be improved. e.g. Figure 3 & 5 are too small and hard to read.

Thank you for this assessment. In an updated version of our manuscript, we will include a more process-based description of our results and discuss implications in greater detail. We have decided to reframe the paper to be more focused on input data, resolution, and snow, while removing in-depth discussion of links of all that on GPP/ET estimates, following valid suggestions from Reviewer 1 and 2. We will further include a more detailed description of snow cover dynamics in CLM5.

We will omit Figure 5 from a revised version of this manuscript but will make Figures 3 easier to read.

---

## Author Response (AR1)

We thank the reviewer for taking the time to read through our paper so carefully, for their detailed review and for their insightful comments. As a result of their suggestions, we have re-focused the manuscript and limited the main analysis to seasonal snow, which lead to extensive rewriting of large parts of the manuscript, the addition of several new figures and an inclusion of new analysis. Please find our replies below as inserted in red text.

**General comments**
The paper by T. Malle et al. addresses the effects of spatial resolution, atmospheric forcing data, and land surface characterisation on snow depth, gross primary production (GPP) and evapotranspiration (ET) simulated by the Community Land Model version 5 (CLM5) over Switzerland. Factorial model experiments with different combinations of resolution and input data were used to isolate the effects of the three sources of variability in the model output. The authors conclude that deficiencies in all three aspects contribute to the current uncertainty in the results of land surface models (LSMs), particularly in heterogeneous regions like the Swiss alps. In this context, they call for the use of more fine-grained input data and for evaluating LSM simulations at high resolution.

The study contains several elements that are interesting for the audience of ESD, including modelers and users of land/climate model output over heterogeneous regions. It probably involved huge efforts, considering the generation of various input datasets, model experiments, the analysis of three simulated quantities (snow depth, GPP, ET), and their comparison against observational or modelled benchmarks. For a high-quality modelling paper, I think the technical foundations need to be improved (e.g. what controls simulated snow depth, GPP, and ET in CLM5-SP offline?). Also, the study addresses several complex issues in land/climate modelling (e.g. ecophysiology, carbon cycle dynamics, feedbacks, etc.) that are poorly captured in the model setup chosen for this study (i.e. CLM5 with prescribed vegetation phenology, largely inactive carbon and nitrogen cycles, and prescribed atmospheric conditions). I think this distracts from the relevant key findings and confuses the overall storyline. Below, I list suggestions how the study and its findings could be improved to become more useful for the audience of ESD. I think this involves substantial revisions. My comments refer to main text only, not the Appendix.

Thank you for this comment, and for the generally positive assessment of our work. Our goal is to maximize the paper's usefulness for the modelling community, so we very much welcomed your suggestions for improvement.

Initially, we opted to use CLM5 with prescribed vegetation phenology to represent the vegetation of Switzerland today in the most accurate manner, which would be very difficult to achieve when running in prognostic biogeochemical mode, at least without data assimilation. We also did/do not have the computational resources to perform CLM5 simulations at high resolution in bgc mode, especially not the required spin-up. We have therefore followed your suggestion of re-focusing the manuscript on snow-related analysis and do not explicitly focus on the link between snow and GPP/ET.

We elaborate in our responses to the specific comments below.

**Specific comments**
**(1) Relevance of higher resolution and implications of the results**. It is obvious that higher resolution and more specific input data improve model performance for a specific location, i.e. when CLM5 output is evaluated against high-resolution simulations or point/site-scale observations. This is known by the modelling community and it is thus not surprising that the 1km simulations with high native-resolution input data perform best. What additional insights can the study provide, e.g. should we focus on (1) high resolution atmospheric forcing data in offline land surface modelling, (2) high resolution land surface data, or (3) high LSM resolution to capture non-linear effects between atmospheric forcing and land surface characterization? Or should model evaluation and benchmarking be done at high resolution, to rule out deficiencies due to insufficient resolution and focus on process uncertainties? What do users get if they run CLM5 at high (<0.5°) resolution with CRU as the atmospheric forcing dataset, is CRU 0.5° simply interpolated and could "smart downscaling" considering the temperature lapse rate be useful as a (built-in) solution? (this would be very interesting for the CLM community!)

We have added discussion of our results in this context in greater detail in the revised version of the manuscript. Our results show that "smart downscaling" based on a high-resolution DEM leads to large model performance improvements with regards to seasonal snow cover. Simple regridding and running at higher spatial resolution did not lead to much in terms of performance gain (see Figure 4 in the revised manuscript). We hope these results, and our updated discussion will be useful to the (CLM5) modelling community.

**(2) Benchmarking and implications**. The purpose of a land surface model like CLM5 is to simulate larger spatial scales ranging from one grid cell to regional domains or the globe, in spatially representative grid cells rather than point fashion (only at very high resolution, grid cells start to resemble point/site-scale conditions…). Therefore, it might be useful to evaluate the output of various simulations (1km, 0.25°, 0.5°) at the lowest resolution of 0.5°, to assess if there are major differences due to input data quality and non-linearities. Such results could inform the modelling community if there is a need to account for small-scale heterogeneity to obtain accurate fluxes and pools at larger spatial scales of one to several kilometres. This might ultimately be more relevant for the purpose of CLM5 (or LSMs in general) than mimicking point/site-scale conditions.

This is a very valid point. We have redone evaluations of gridded snow simulations at 0.25°, as we believe that given the complexity of the topography across our modelling domain and its relatively small size, and considering today's ever-increasing computational resources, 0.25° should be a fair target for the main analysis.

We have upscaled results from the 1km simulations to 0.25°, using a conservative upscaling approach which preserves areal averages. For this purpose, we had to decrease our evaluation domain slightly, as we performed the 1km simulations with a mask running exactly along the edges of our modelling domain, making it impossible to upscale these areas to 0.25° without crude assumptions. The 0.5° simulations were downscaled to 0.25°, and all simulations were evaluated across the same domain.

As seen from the updated version of the Taylor's diagram (Figure 4 in the revised manuscript / also printed below for convenience), the difference between different land-use datasets with regards to simulated snow depth remains small and increasing spatial resolution in isolation only has a marginal effect on accuracy of simulated seasonal snow cover. Upscaled 1km simulations with highest quality meteorological forcing datasets (Clim$_{OSHD1km}$) perform best during all three points in time (early accumulation, mid accumulation, ablation period); performance of Clim$_{CRU}$ is substantially improved when running with the lapse-rate corrected meteorological forcing (Clim$_{CRU*}$), underlining the effect of a relatively simple lapse-rate based downscaled temperature input to better account for sub grid variability. We have discussed these results in greater detail in the updated version of the manuscript.

[Figure]

**(3) Land surface dataset methodology and evaluation**. The generation of a high-resolution land surface dataset based on national land cover data and satellite-based LAI (not sure I interpreted the brief description in section 2.3.2 correctly) seems quite innovative and interesting. I think the procedures should be described in more detail, so that other could potentially follow a similar approach. Some summary/evaluation/validation beyond what is shown in Figure 1 (e.g. how did land cover fractions and LAI change across Switzerland, regionally, or for the point locations?).

We have described how the land surface dataset was created in a more detailed manner, please read through section 2.3.2. of the revised version of the manuscript. Additionally, we have added additional figures to the appendix showing land-unit fractions (e.g., crop, vegetation), PFT-distributions as well as LAI/SAI spatiotemporally across the model domain for both the high-resolution and the global-based dataset (see Appendix B). Figure 6a of the revised manuscript further shows a direct comparison in overall PAI (averaged across all PFTs as well as between January and March) between the LU$_{HR1km}$ and LU$_{Gl1km}$ dataset.

**(4) Overall scope including snow depth and GPP/ET**. The manuscript might benefit from limiting the scope and analysis to snow depth, and in that area developing the causes for improved model performance more thoroughly (e.g. how can spatial resolution, atmospheric forcing data, and land surface characterization influence snow depth considering forced precipitation, rain/snow partitioning inside CLM5, land cover, LAI, slope, etc.). A caveat of this is, of course, that snow depth is likely closely linked to the forced precipitation and temperature fields, which could make the results appear trivial. Yet, the authors might identify interesting aspects related to, e.g., land cover, LAI or slope. In any case, there are multiple reasons why the

analysis of effects on GPP and ET should be excluded, or included only after substantial revisions:

a) The link between Hypothesis 2 and the chosen methods is currently very weak. Most importantly, the modelling setup and the correlation analysis do not allow to isolate the effects of differences in snow depth from differences in its drivers (i.e. spatial resolution, atmospheric forcing data, and land surface characterization) on GPP and ET. The framing of GPP and ET as "snow-cover dependent ecophysiological variables" is confusing, because likely most of the GPP and ET differences are driven by the resolution-dependent forcing fields directly (i.e. atmospheric variables or land surface characterization) and not indirectly via snow-cover changes. This joint independent driver could lead exactly to the correlation between snow cover and GPP/ET found by the authors, without any effect of snow cover itself on GPP/ET.

b) In CLM5 in SP mode, vegetation phenology is prescribed as a climatological seasonal cycle of LAI, and LAI controls the leaf to canopy scaling of all fluxes including carbon (GPP) and water (ET). Therefore, in SP mode the model has very limited capabilities to show an ecophysiological response to snow cover change, in the sense of seasonally shifting growth. As a minimum, I recommend to discuss the precise implications (e.g. is the response we see solely due to changes in temperature and water availability, or what can affect GPP in SP mode at all?). However, I strongly doubt that an ecophysiological response can be quantified in SP mode. I imagine it works as follows, although I am not 100% sure: if the snow season is shorter than in the climatology, suitable temperatures for growth coincide with zero LAI in the model and nothing happens until the climatological growth begins; if the snow season is longer than in the climatology, suitable temperatures for growth coincide with high LAI in the model, leading to a jump start and no compensation later in the season.

c) Ecophysiology refers specifically to (plant) organisms and not to land as a whole. So, for ecophysiological effects on ET I suggest analyzing plant transpiration and canopy evaporation fields of CLM5, or total ET in the vegetated parts of grid cells excluding the bare soil PFT. However, considering that the experiments include changes in land surface characterisation, the best option might be to remove the term "ecophysiological" and to refer to responses in land ET, which include e.g. the effects of varying bare soil fraction.

d) Simulated GPP and ET are compared against the "best-effort" configuration of CLM5 at 1km resolution as a benchmark, and deviations from this benchmark are considered model uncertainty. An objective benchmark (e.g. observational data or output of a dedicated model, like for snow depth) is lacking. Also, I am not sure the term "model uncertainty" is appropriate in this context. Is it model uncertainty if lower resolution models perform poorer at producing high-resolution-model-like outputs? The comparison could potentially be done the other direction (see point 2).

Thanks for the suggestions and elaborate explanations. We have followed the suggestion of re-focusing the manuscript on snow depth and did not focus on the link between snow and GPP/ET. We have discussed rain/snow partitioning in more detail (see Section 2.1.2 "Rain-snow partitioning in CLM5") and included an additional analysis of snow-depth accuracy per elevational band (Figure 5). For the point-scale simulations we have added an extra simulation which matches station locations from a land-use perspective (open land, no forest = PFT0 / bare ground).

The more comprehensive comparisons of point-scale model simulations to observations of snow depth (wiggle plots, Figure A1 in the original manuscript) was moved to the main paper (now Figure 3). We have kept the analysis of GPP/ET in the paper, however, not as a main result, but in order to demonstrate how different GPP/ET results can be in our various simulation setups. We have further added a new analysis to better understand why the effect of land-use data in our

results was minimal. Figure 6 shows that the majority of snow-dominated pixels correspond to pixels with little change in PAI between the high-resolution and the global land-use datasets (e.g. non-forested areas), giving context to our results. The low sensitivity we find with regards to land-use forcing hence has to viewed critically, as it is mostly a symptom of the properties of the snow regime in our model domain.

**(5) Introduction**: The background could be more technical (e.g. what controls simulated snow depth, GPP, and ET in CLM5-SP offline?) and focused on aspects that matter for this study (i.e. strengthen the main story and cut out things that are interesting but not directly relevant). We have revised both introduction and methods to include more technical background with regards to snow modeling. There is now a dedicated section in the methods giving technical background to snow CLM5 simulations for readers convenience (see Section 2.1.1 - Snow and fractional snow cover schemes in CLM5).

**(6) Discussion**: The findings need to be contextualised, considering the capabilities of CLM5 in SP mode offline (see above). For the comparison with other studies (e.g. Birch et al.) to be valid and useful for the reader, it would be good to understand what these studies did or how they explain their biases (e.g. also land cover specification, atmospheric forcing and model resolution, or something completely different like effects of grazing animals or artic plant types?). Certain links made between the findings of this study (with a very specific setup and research focus) and model uncertainties (with sometimes known different root causes) are not appropriate (e.g. L371). I recommend reconsidering these and focusing the discussion more on new insights gained through this study (see points 1 and 2). We have revised the discussion and focused it more on the highlights of this study.

**(7) Language and figures**: The manuscript is very well written. There are a couple of "empty" phrases that highlight something but do not actually deliver new insights (e.g. the results have profound implications, the study highlights the importance of model development, the study highlights the utility of multi-resolution modelling, etc.). I think those could be filled with content or removed. The figures have high quality and are visually appealing. Thank you. We have removed empty phrases from the paper.

**Technical and line-by-line comments**
L14 ff: "Earth's systems" sounds a bit unconventional and far-reaching; maybe use a more concrete/narrow term? We have rephrased this sentence, it now reads: *'By embracing high-resolution modeling, we can enhance our understanding of the land surface and its response to climate change.'*

L19: add water Done.

L23, L24: maybe use "influence" instead of "control/determine" Done.

L24, L35, L65: there is no feedback among the mentioned dependencies/effects We have omitted large part of this paragraph in the revised introduction and used 'exchange' instead for the remaining part.

L29, L31: check logical link for "thus" and "as" Done.

L49: offer literature for multi-resolution modelling?
We have cited Singh et al. (2021, https://doi.org/10.1002/2014WR015686) and Meissner et al. (2009, https://doi.org/10.1127/0941-2948/2009/0400 ) here.

L50: they allow evaluating Done.

L51: I guess any gridded LSM can be evaluated against point/site-scale observations by taking the individual grid cells that match the locations best; there is no need for dedicated point simulations for this
In principle yes, but for this study we used downscaled meteorological forcing data to the exact location of the point observation for our '1km' simulations, which in case of the OSHD dataset also included station observations and should represent a best-case scenario. Also, for the lapse-rate based temperature correction we used the exact coordinate and a high-quality GPS elevation measurement for the downscaling. We have tried to make this clearer in the methods section of the revised manuscript (see Section 2.2, 2.3.1). We also acknowledge that our choice of naming of the point simulations might have fueled this confusion, which we have therefore updated in a revised version of this work (e.g. simply $Clim_{OSHD}$ and $Clim_{CRU*}$ instead of $Clim_{OSHD1km}$ and $Clim_{CRU*1km}$) to make this clearer.

L56: what is meant by snow cover "dynamics"? the temporal evolution?
Here we mean depth, duration as well as variability of snow cover across space and time. We have updated this sentence accordingly to make this point clearer.

L73: consider limitations in SP mode ok

L80: in my understanding, "process representation" refers literally to how processes are represented in the model, i.e. the equations used to calculate snow depth, GPP and ET; I think this term is not appropriate for the modifications in forcing/input data and resolution made in this study
We have changed this part of the introduction, and avoided using the expression 'process representations'.

L87: remove "heat fluxes" if not addressed in the results
Heat fluxes were removed here.

L88-90: consider reformulating, the sentence sounds nice but it does not deliver any content (or at least I cannot understand it)
We have reformulated the sentence. The last part of our introduction now reads:
*Our findings can inform the optimal design of further offline applications of LSMs, for instance 1) to extrapolate local-scale experimental findings; 2) to address the limitations of global-scale, coarse resolution simulations; and 3) to support the interpretation of snow cover information contained in Earth System simulations.*

L98-102: the methods need some technical precision here: which state variables, which datasets (e.g. what time period/ past conditions do they represent), not only "natural" vegetation (crop

PFTs), how does GPP work in SP mode (if the LUNA model active, also cite Ali et al. 2016 for photosynthesis)

We have cited Ali et al. 2016 (LUNA model is active in our simulations), and give a more detailed background with regards to the land-use datasets used in Section 2.3.2. As we have shifted the focus of the paper to mostly snow-related analysis we did not give additional background on GPP computations in CLM5.

L100: "is approximated": I think this a used choice, not done by the model.

We have rewritten this sentence, it now reads:

*GPP for the context of this study was approximated by photosynthetic activity, with photosynthesis being limited by carboxylation, light, and export limitations for different plant functional types*

L103: revise components of ET, e.g. soil sublimation does not make sense; maybe "ice" is missing?

Yes, soil should read ice here, we have updated it accordingly.

L105-111: snow cover is the focus of this study, so I think the foundations of snow cover calculations should be provided for convenience (and understanding), including rain/snow partitioning in CLM5

We have documented and explained CLM5 snow cover calculations including rain/snow partitioning in greater detail – please have a look at Section 2.1.1 and 2.1.2.

L115, L197: PFTs are patch level, mention that prescribed LAI etc. is in SP mode.

We have updated these sentences accordingly.

L116-L119: is there any difference between taking out individual grid cells from the regional simulations and running dedicated point simulations (i.e. a 1x1 regional grid) in your setup? they should be identical, provided that (1) the grid is anchored and specified identically, and (2) there are no lateral exchanges between grid cells (which depends on the CLM5 compset, if river routing is off there is no lateral exchange)

There is no lateral exchange in our model setup, so in principle there is no difference in the simulation setup. However, as mentioned above we used downscaled meteorological forcing data to the exact location of the point observation, resulting in different forcing datasets than for the 1km simulations. We have tried to make this clearer in the text.

L124: for comparing between resolutions, it would have made sense to subdivide the 0.5° and 0.25° grids, e.g. by using 4x the number of cells for 2x resolution, i.e. 20x12 grid cells for 0.25°; that way you could preserve the grid anchoring and preclude differences due to a new "positioning" of the grid; maybe motivate your choice and/or mention potential effects of different grids on the results

The 1km grid was pre-determined by the OSHD grid of snow-simulations and meteorological forcing data, which we used as a starting point. The 0.5° and 0.25° grid were then determined to closely match the extent of this initial grid. As a result grid anchoring might slightly vary between resolutions – we have added a comment regarding this caveat to the methods section.

*('As the 0.5° and 0.25° grids were chosen to closely match the extent of the pre-determined 1km grid, grid anchoring might slightly vary between resolutions.')*

L128: a "the" is missing
Thanks for catching this, "on one hand" was changed to "on the one hand".

L137 ff: are the grid cells for point simulations centered on the station coordinates? using the "nearest neighbor" grid cell for land surface characterization seems like a very simplified approach compared to all the other sophisticated things done in this study; why not use conservative regridding so you would get something more realistic? ideally, one would generate surface data from the raw PFT fraction data at 0.05° resolution (CLM's own methods) or the raw Swiss national data (your methods); in contrast, taking the nearest neighbor effectively shifts surface information and pairs it with the correctly positioned atmosphere; if this is acceptable, one might as well take the nearest (or interpolated/regridded) results of the gridded simulation? (see comment on L116-L119)
Yes, the grid cells for point simulations are centered on the station coordinates. We did actually first calculate the domain and surface dataset for each point location separately via CLM's own methods (not with the raw PFT fraction data at 0.05° but at 0.25° though). We then updated this dataset with the 1km HighRes dataset accordingly.
We acknowledge that ideally we would have re-run our own methods for each station location, but did not have the resources to do so. However, since all snow station locations are open, non-forested sites we prepared additional simulations which ensure that we have a 100% vegetated non-forested grid cells. We have included these new simulations in Figure 2 (LU$_{nofor}$) and described it accordingly in the text. We also acknowledge that we should re-name the point simulations (e.g. simply LU$_{HR}$ and LU$_{Gl}$ instead of LU$_{HR1km}$ and LU$_{Gl1kml}$) to avoid confusion and make this point clearer.

L154: is "accelerated decomposition" valid/applicable for SP mode? it sounds like BGC; by "cycling" (remove "re-")
You are right, of course "accelerated decomposition" is not necessary for SP mode, this part of the sentence was removed. "Re-cycling" was replaced by "cycling".

Figure 1 caption: is "percentage vegetation cover" the natural vegetation landunit including bare soil, or the sum of vegetation PFTs and CFTs? (the latter would be good)
Figure 1 of the revised manuscript now shows the sum of vegetation PFTs and CFTs (only marginal differences to the original figure which showed the natural vegetation landunit including bare soil).

L159, L177: is CRU a station-based interpolated dataset and the OSHD based on the COSMO model? for OSHD, it is also a bit unclear if the dataset was produced or re-used for this study
Yes, we have made this clearer in the text of the revised manuscript. The OSHD dataset is used operationally by the snow hydrological service, it was hence mostly re-used for this study.

L187: was the native 0.05° PFT and LAI data reprojected and regridded, or was this done based on an existing surface dataset at e.g. 0.5° resolution? depending on the data used, there might be several regridding steps involved (with every step further degenerating the final product) and the native input might be 0.05 or 0.25°; was this done with the CLM5 tools, with which regridding algorithm (bilinear, conservative)?

CLM5 tools were used separately for the 1km, 0.25° and 0.5° resolution to generate each of the 'global' datasets of this study (LU$_{Gl0.5}$, LU$_{Gl0.25}$, LU$_{Gl1km}$), using the conservative regridding algorithm and all underlying raw input data files. However, as mentioned above the native input of the underlying raw PFT data used was 0.25°.

L200: FSM2 output is not "observational"
We have clarified this in the revised manuscript.
L216-217: remove "were", "ground truth" is usually used in remote sensing?, does "upscaled" mean regridded and if so, with which algorithm (bilinear, conservative)?
Ok - "were" will be removed, and "ground truth" will be replaced by "best-case reference". Here upscaled means regridded with the conservative algorithm. We will add this information to the sentence.

L230-234: consider doing this at 0.5° resolution, see point 2
See our answer to point 2 earlier.

L238: to be honest I am a bit lost by now – was 3.1.1 done with the point simulation results? maybe for each Results sub-section this could be highlighted in the title or mentioned in the first sentence.
Thanks for this note, we have reduced the number of headers to make it easier to follow. The results section of the revised manuscript now includes the following headers:
      3.1. Evaluation of snow simulations at point locations
           3.1.1.  Accuracy of FSM2 point-scale simulations
      3.2. Evaluation of gridded snow simulations
      3.3. Simulation of ecophysiological variables

L247: for this section it would be really good to understand how land cover and LAI and potentially affect snow depth in CLM5 (see point 4)
Please look at Figure 6 in the revised manuscript, where we compare changes in LAI between our two land-use datasets and link it to snow height. We demonstrate that for pixels with a lot of snow cover, there is little change in LAI. This further gives context to the low sensitivity of our simulations with regards to land-use forcing dataset.

L283: replace "parameters" by "variables" Done.

L284: why is peak GPP assessed and not total GPP? I see the "motivation" later in L328, but because there is a bigger effect does not mean it is more relevant? I think this is related to limitations in SP mode (see point 4b); the effects described in section 3.2 go way beyond ecophysiology (see point 4c)
We have update Figure 4 (now Figure 7 in the revised manuscript) to use show total annual GPP instead of peak GPP, and also rewritten parts of the results section here.

Figure 4 labels: replace "climatological" by "meteorological" Done.

L322: snow cover and ET are negatively correlated, but I doubt this is driven by snow but rather by cold temperatures and energy (not water) limitation (see point 4a); feedbacks to the atmosphere are missing in CLM5 offline by construction

Following your suggestion in the specific comments we have refrained from discussing these links in the paper. The section 'Seasonal snow cover development and ecophysiological variables' was completely omitted from the paper. Instead, we have prepared a more detailed assessment of snow-related simulation results as explained above.

L348 ff: for the calculation of variations in (monthly) total GPP across Switzerland, it would be useful to have an observational benchmark and to relate the amounts to total GPP (i.e. % variation of total GPP). Is there a good reason for not calculating variation in total annual GPP? I would find this quantity more informative

This part of the discussion has been deleted in the revised manuscript. As we shifted this paper's focus, we have not included observational benchmarks for GPP but discuss relative differences.

L368: ET can also be water limited in Switzerland, at least in some regions seasonally

Yes, thanks for that note. As we are not focusing on links between snow and ET/GPP in the revised version of the manuscript we have omitted this section of the discussion from the revised version of the manuscript.

We thank the reviewer for taking the time to read through our paper, for their detailed review and for their insightful comments. As a result of their suggestions, as well as suggestions of Reviewer 1 we have re-focused the manuscript and limited the main analysis to seasonal snow, which lead to extensive rewriting of large parts of the manuscript, the addition of several new figures and an inclusion of new analysis. Please find our replies below as inserted in red text.

Although land surface modeling has evolved from simple biophysical parameterizations to complex frameworks in recent years, large uncertainties remain especially in mountainous regions and areas with complex terrain. This study uses a multi-resolution modeling setup to investigate the impact of meteorological forcing data, spatial resolution, and land surface data on the simulation of snow cover and ecophysiological variables. The authors perform simulations using the Community Land Model version 5 (CLM5) over Switzerland. They found that increased resolution not only improved the representation of snow cover in CLM5 but also propagated through the model and affecting the gross primary productivity (GPP) and evapotranspiration (ET).

Overall the manuscript is well written and of interest to the land surface/earth system modeling community. However, the CLM5 model setup and the model evaluation method do not appear to be appropriate in the current manuscript. In specific, (i) the CLM5 is setup in prescribed satellite phenology mode (~ fixed growing season), yet one of the main focus of the study is to investigate the link between snow cover duration and growing season length (and GPP/ET); (ii) CLM5 simulations in the study were conducted at three resolutions: 1km, 0.25deg, and 0.5deg, but the evaluations were performed at 1km resolution (Figure 3), which are not fair comparisons in my opinion.

I suggest the authors choose the prognostic biogeochemistry mode for CLM5 simulations, and perform model evaluation at the resolutions of the respective simulations. In addition, I have some minor suggestions outlined in the comments below that will hopefully improve the future version of the paper.

Thank you for this comment.

**Regarding (i):** Initially, we opted to use CLM5 with prescribed vegetation phenology to represent the vegetation of Switzerland today in the most accurate manner, which would be very difficult to achieve when running in prognostic biogeochemical mode, at least without data assimilation. We also did/do not have the computational resources to perform CLM5 simulations at high resolution in bgc mode, especially not the required spin-up. We acknowledge the problems arising from this setup when focusing on the link between snow cover and growing season length and have decided to reframe the paper following a valid suggestion from Reviewer 1. The revised manuscript is more focused on input data, resolution, and snow, while in-depth discussions of links of all that on GPP/ET estimates is removed. For this purpose, we have moved Figure A1 from the appendix to the main paper (now Figure 3), and further include more detailed performance assessments of the various CLM5 model setups along elevational bands (see Figure 5 of the revised manuscript). We further include an additional analysis on links between change in land-use and simulated snow cover (see Figure 6 of the revised manuscript).

**Regarding (ii):** This is a very valid point. We have redone evaluations of gridded snow simulations at 0.25°, as we believe that given the complexity of the topography across our

modelling domain and its relatively small size, and considering today's ever-increasing computational resources, 0.25° should be a fair target for the main analysis.
As seen from the updated version of the Taylor's diagram below (Figure 4 in the revised manuscript), the difference between different land-use datasets with regards to simulated snow depth remains small and increasing spatial resolution in isolation only has a marginal effect on accuracy of simulated seasonal snow cover. Upscaled 1km simulations with highest quality meteorological forcing datasets (Clim$_{OSHD1km}$) perform best during all three points in time (early accumulation, mid accumulation, ablation period); performance of Clim$_{CRU}$ is substantially improved when running with the lapse-rate corrected meteorological forcing (Clim$_{CRU*}$), underlining the effect of a relatively simple lapse-rate based downscaled temperature input to better account for sub grid variability. See Section 3.2. for a more in-depth description of these results.

[Figure]

Specific comments

L164, the sentence does not read well.
We have reformulated this sentence, it now reads:

*We selected CRU-JRA due to its large timespan (1901-2020), which includes recent years and hence ensures sufficient overlap with our high-resolution forcing dataset (see below), as well as due to its application in the annual Global Carbon Budget assessments (e.g., TRENDY, Friedlingstein et al. (2020)) and in the Land Surface, Snow and Soil Moisture Model Intercomparison Project (LS3MIP, Hurk et al. (2016)).*

L169-170, do you assume values in the original dataset (ClimCRU1km) are at sea level, and apply the temperature lapse rate based on the mean elevation of each 1km grid from the CRUJRA data? Which elevation data do you use? I would suggest include a map of elevations in Figure 1.
We do not assume that the values in the original dataset are at sea level. Rather, we use a global DEM at 0.5deg to first bring temperature to sea-level temperatures by applying negative lapse rates. We then use a high-resolution DEM of Switzerland at 1km to re-lapse temperature. We have added this information to Section 2.3.1 of the revised manuscript. We have also included a map of both the high and low resolution DEM (see Figure C1 in the appendix of the revised manuscript).

L172, it would be helpful to add a description of the snow/rain partitioning method in CLM5.
CLM5 partitions total precipitation into rain and snow according to a linear temperature ramp, resulting in all snow below 0 °C, all rain above 2 °C, and a mix of rain and snow for intermediate temperatures. We have dedicated a section in the revised manuscript to this, see Section 2.1.2 ("Rain-snow partioning in CLM5") of the revised manuscript.

L181-182, do you just aggregate the 1km data to 0.25deg and 0.5deg? Please try to describe what exactly is being done. The ClimOSHD forcing data would be useful for other modelers, is the data available?
Yes, here the 1km data was upscaled to 0.25 and 0.5deg using a conservative regridding approach. We have expanded this methods section to describe what exactly was done more comprehensively.
Thanks for your note, we will consider publishing the Clim$_{OSHD}$ forcing data, but certainly as a separate data paper given the enormous effort required to prepare such datasets
Simulation results and updated surface data from this study will be made available as part of this paper.

L234, given that the met forcing and landcover data etc. are all at coarser resolutions, it is not fair to evaluate coarser resolution (0.25deg, 0.5deg) CLM5 simulations using finer resolution (1km) observations. I suggest the authors regridding the 1km observation data to the 0.25deg and 0.5deg first, then redo the comparisons and Figure 3.
As mentioned in our comment above, we have followed your advice and redone the evaluation at 0.25deg, which we believe is a fair target. We have updated the Taylor diagram and results accordingly.

L241-245, it would be helpful to show or discuss which variables in the met forcing data contribute to the different CLM5 simulations.
We have included more detailed maps of differences in temperature and precipitation between the different forcing datasets (see Appendix C of the revised manuscript), and further discuss contributions to snow simulations in more detail.

Figure 3 is an important figure in the paper, but the Taylor plots and labels/legend are too small, and hard to read.
Thanks for pointing this out. We have improved the quality of the Taylor plots, hopefully making them easier to read.

In the captions of all the figures, a summary of main results is also included, which is not necessary and makes the captions too long.
We have reduced the captions of most figures, and refrained from giving a summary of our results in the figure captions.

L256, supplementary material is not found.
This was a mistake and should read Figure A1, thanks for spotting it. Figure A1 will be moved to the main manuscript in a revised version of the manuscript.

L265, Figure 3 needs to be cited here.

Yes, thanks for pointing this out. We have cited Figure 3 here in the revised version of the manuscript.

L273-275, I suggest the authors redo these evaluations at the resolutions used for each CLM5 simulations.
We have redone all evaluations at 0.25deg, which we believe is a fair target for our model analysis.

L290-291, the sentence doesn't read well.
We have reformulated this sentence, it now reads:
*For GPP, effects of land-use information outweighed effects of meteorological forcing. Higher level of detail in the land use data caused both increases and decreases in GPP across the model domain, while improved meteorological input had a more systematic effect.*

Figure 4, note the 3$^{rd}$ panel are labeled as effect of climatological forcing instead of meteorological forcing.
Thanks for catching this, we have updated the label.

L316-317, the sentence does not read well.
We have omitted this section from the revised version of the manuscript.

Reviewer # 3

We thank the reviewer for taking the time to read through our paper and for their insightful comments. As a result of their suggestions, as well as suggestions of Reviewer 1 and 2 we have re-focused the manuscript and limited the main analysis to seasonal snow, which lead to extensive rewriting of large parts of the manuscript, the addition of several new figures and an inclusion of new analysis. Please find our replies below as inserted in red text.

The manuscript by Malle et al. investigates the impact of spatial resolution, quality of atmospheric forcing datasets and land-use information on the simulated snow depth, GPP and ET over the spatial extent of Switzerland and adjacent watersheds of neighboring countries by using the Community Land Model 5 (CLM5). Simulations of different combinations of meteorological forcing and land-use information were conducted to analyze changes in model performance. In addition, CLM5 simulated snow depth were compared with station observations and results from a spatially distributed, physics-based snow model. The authors find the combination of increased spatial resolution of model and high-quality input datasets can improve the representation of snow cover in CLM5, and these improvements further propagate through the model, directly affecting GPP and ET. The manuscript demonstrates the importance of high spatial resolution and high quality input datasets for climate impact studies.

The manuscript dedicated a detailed description of methodology, but the explanation of the results is somewhat brief, and most of them are descriptive, lacking of model processes related analysis and discussion. Such as, what controls the snow depth simulation in CLM5, how the different forcing datasets affect snow simulation? how the improvements in snow propagate in CLM5 in a cascade way, what's the linkage between snow cover and GPP and ET. I suggest the authors improve these parts. In addition, the figures in the manuscript should be improved. e.g. Figure 3 & 5 are too small and hard to read.

Thank you for this assessment. In the revised version of our manuscript, we have included a more in-depth description of our results and discuss implications in greater detail. We have reframed the paper to be more focused on input data, resolution, and snow, while removing in-depth discussion of links of all that on GPP/ET estimates, following valid suggestions from Reviewer 1 and 2. We have further included a more detailed description of snow cover dynamics in CLM5 (see Section 2.1.1. "Snow and fractional snow cover schemes in CLM5" of the revised manuscript). We have omitted Figure 5 from a revised version of this manuscript but have increased the size of the Taylor diagram (original Figures 3, now Figure 4) and made it easier to read.

---

## Author Response (AR2)

We thank the editor and the reviewers for taking the time to read through our revised paper and for their positive assessment of our work. Please find our replies to the points of reviewer 2 below as inserted in red text.

I think the authors did an excellent job revising the manuscript by focusing on snow-related analysis. I only have some minor comments as outlined below. Thank you.

One of the main findings in this study is "Results demonstrate the need for resolutions higher than 0.25deg for accurate snow simulations in topographically complex terrain (L10)". One question one may ask is: Does the resolution have to be 1km? Would 5km or 10km resolutions work? It'd be nice if the authors could address this question by performing some additional tests. While we agree that this is an interesting question, this would require significant additional computational resources. Unfortunately, we are not able to perform such tests at this stage and will reserve it for future studies.

L12 change "recling" to declining. Done

L154, note Wmax is the maximum accumulated snow water equivalent. Done

Eq (3) in Swenson and Lawrence (2012) is incorrect. According to CLM5 code, Nmelt = 200/max(10, σtopo). Thanks for catching this, we corrected the equation.

Section 2.4.2, I would suggest including a brief description about the physics in FSM2, e.g. what's the snow and fractional snow cover schemes? Rain-snow partitioning method? We have included a brief description of the physics in FSM2, please have a read through Section 2.4.2.

L360-361: I'd suggest rewording "too fast settling". Done

Table A1, some information about the snow depth would be helpful, e.g. the maximum snow depth for each station. Thanks for this suggestion. We have included an extra column showing maximum measured snow depth for each station during the 2017/18 season.

Fig.C2&C4 show that ClimCRU tends to have more precipitation than ClimOSHD in the southern part of the study area, can you explain and/or elaborate this a bit? The reason for higher precipitation rates of $Clim_{CRU}$ in the southern part of the study area (Ticino) most likely is due to the inability of the relatively coarse resolution CRU dataset to reflect topography, which ultimately is responsible for the dryer conditions in the inner alpine valleys in the south. Instead, precipitation patterns in the south of the $Clim_{CRU}$ dataset are properly affected by the drier conditions of the Po plain. At the same time, $Clim_{OSHD}$ might slightly underestimate precipitation rates in this area, as a larger proportion is falling as rain as compared to the northern areas and rainfall is not corrected since assimilation of snow data can only correct for biases in snowfall.